# G6PD and ACSL3 are synthetic lethal partners of NF2 in Schwann cells

Athena Kyrkou[1,2], Robert Valla[1,2], Yao Zhang [1,2], Giulia Ambrosi[3], Stephanie Laier[4], Karin Müller-Decker[4], Michael Boutros[2,3] & Aurelio A. Teleman [1,2] ✉

Neurofibromatosis Type II (NFII) is a genetic condition caused by loss of the *NF2* gene, resulting in activation of the YAP/TAZ pathway and recurrent Schwann cell tumors, as well as meningiomas and ependymomas. Unfortunately, few pharmacological options are available for NFII. Here, we undertake a genome-wide CRISPR/Cas9 screen to search for synthetic-lethal genes that, when inhibited, cause death of NF2 mutant Schwann cells but not NF2 wildtype cells. We identify *ACSL3* and *G6PD* as two synthetic-lethal partners for *NF2*, both involved in lipid biogenesis and cellular redox. We find that *NF2* mutant Schwann cells are more oxidized than control cells, in part due to reduced expression of genes involved in NADPH generation such as *ME1*. Since G6PD and ME1 redundantly generate cytosolic NADPH, lack of either one is compatible with cell viability, but not down-regulation of both. Since genetic deficiency for *G6PD* is tolerated in the human population, G6PD could be a good pharmacological target for NFII.

Neurofibromatosis Type II (NFII) is a genetic condition characterized by a predisposition to benign tumors of the peripheral nervous system[1]. It is caused by homozygous loss-of-function of the homonymous gene *NF2*[2,3] in Schwann cells, the glia that ensheath peripheral nerves. Individuals with NFII typically harbor a germline mutation in one of the two *NF2* alleles. Recurrent, stochastic, somatic loss of the second *NF2* allele gives rise to tumors throughout the lifetime of the patients[1]. These tumors grow and constrict the nerves. Since cranial nerves are often affected, this leads to balance problems, dizziness, headaches, facial weakness, and loss of hearing or sight. There are few pharmacological options for treating NFII[4,5] although bevacizumab has recently shown promise[6]. Tumors are surgically resected, leading to nerve damage. Since patients recurrently develop new tumors, this results in significant morbidity.

NF2 is an upstream regulator of the Hippo/YAP pathway[7,8]. When *NF2* is mutated, the Hippo tumor suppressor pathway is inactivated, leading to activation of the downstream oncogenic transcription factors YAP and TAZ. This pathway is difficult to target pharmacologically because most of the upstream pathway components are tumor suppressors, and hence already inactive in the tumor condition, and the downstream effector components YAP/TAZ are transcriptional coactivators, which are difficult to target with small molecules. Recent work has focused on the development of inhibitors targeting the TEAD transcription factors, which recruit YAP/TAZ to chromatin via their DNA-binding domains[7–9]. This approach is very promising[10], although toxicity may arise, given that genetic loss of *YAP/TAZ* may lead to impaired maintenance of Schwann cell myelination or to impaired remyelination after injury, and consequently impaired nerve function[11–14].

We undertook here a synthetic lethality approach to search for genes that only cause cell death when they are inactivated in combination with NF2 loss-of-function. Pharmacological inhibition of these gene products should therefore specifically cause death of NF2 tumors. This identified G6PD (Glucose-6-phosphate dehydrogenase) and ACSL3 (Acyl-CoA Synthetase Long-Chain Family Member 3) as potential therapeutic targets for NFII.

[1]German Cancer Research Center (DKFZ), Division B140, 69120 Heidelberg, Germany. [2]Heidelberg University, Institute of Human Genetics, 69120 Heidelberg, Germany. [3]German Cancer Research Center (DKFZ), Div. Signaling and Functional Genomics, 69120 Heidelberg, Germany. [4]Core Facility Tumor Models, German Cancer Research Center (DKFZ), Heidelberg, Germany. ✉e-mail: a.teleman@dkfz.de

## Results

### Generation of an isogenic pair of NF2-WT and NF2-KO human Schwann cells

We aimed to identify genes that cause cell lethality when inhibited in *NF2* mutant Schwann cells but do not cause lethality when inhibited in wildtype cells. To this end, we first generated a pair of isogenic Schwann cell lines that differ only by their *NF2* status. Starting with the

human Schwann cell line ipn02.3 2λ, which is a wildtype line ("WT") derived from normal human nerve, and therefore expresses NF2[15] (Fig. 1A), we knocked-out *NF2* by CRISPR/Cas9 using two independent sgRNAs that target its first two coding exons, respectively. This yielded two independent lines containing *NF2* frameshift mutations, which should lead to premature stop codons, and very short, truncated NF2 peptides (Suppl. Fig. 1A). These knockout cells have no detectable NF2

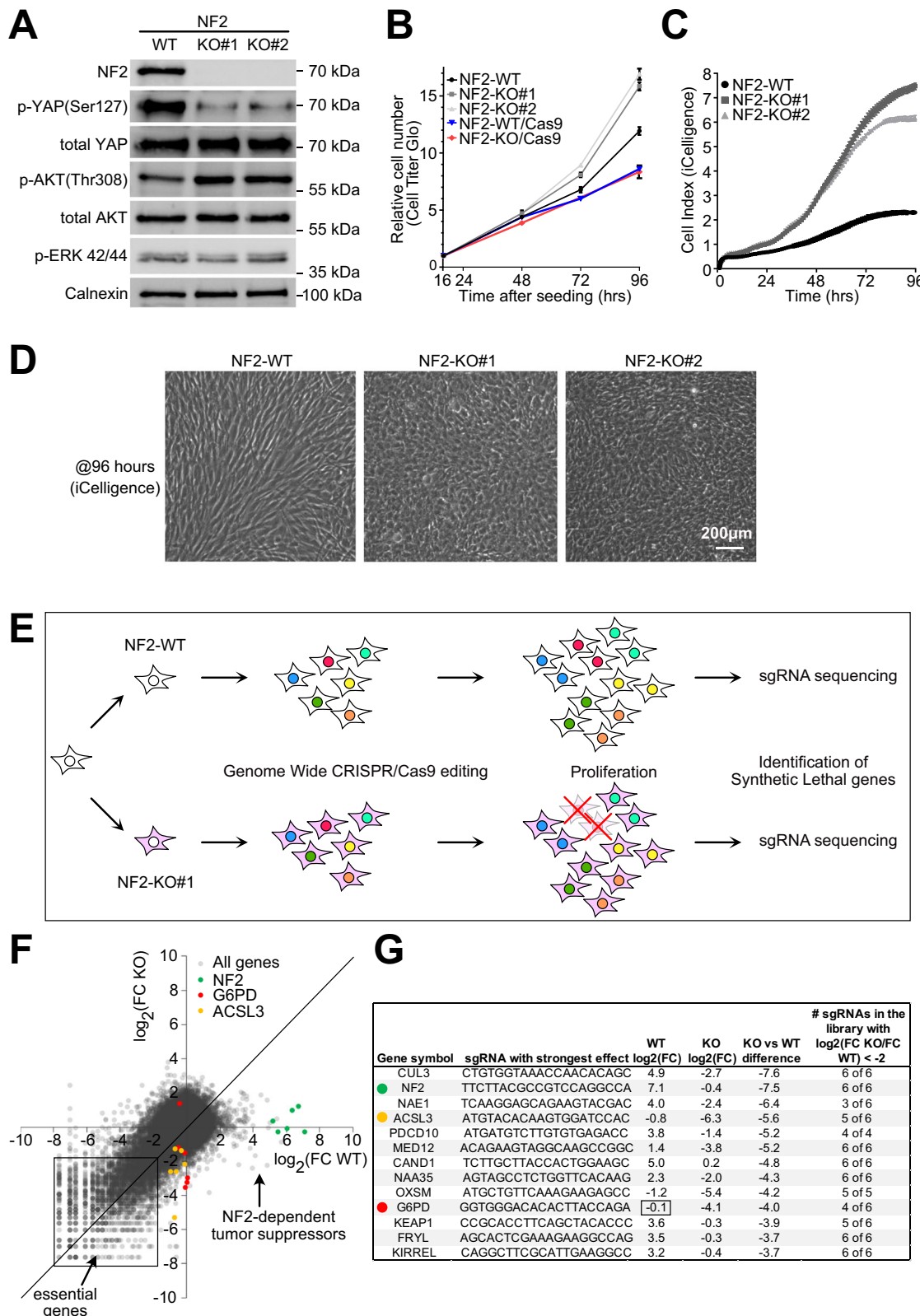

**Fig. 1 | Synthetic lethality screen identifies G6PD and ACSL3 as NF2-synthetic-lethal genes. A** NF2 knockout (NF2-KO) cells have reduced YAP phosphorylation and elevated Akt phosphorylation. Western blot analysis of lysates from isogenic NF2 WT and KO human Schwann cell lines. pERK 42/44: pThr 202/pTyr204. **B** At low cell density (2500 cells/0.32cm²), NF2-KO cells proliferate at roughly the same rate as NF2-WT cells. Relative cell number measured by CellTiter-Glo. Results are shown as a fold-change relative to the first timepoint (16 hrs postseeding). $n = 4$ wells. Bars = mean ± SD. **C** NF2-KO cells are not contact inhibited compared to NF2 WT cells. Real-time, impedance-based (iCelligence) cell growth analysis of NF2 WT and KO isogenic Schwann cells starting at medium confluence (5000 cells/0.32cm²). $n = 4$ wells. Bars = mean ± SD. **D** At high confluence, NF2 KO Schwann cells are more packed and less elongated compared to NF2 WT cells. Images taken at full confluence (96 hrs timepoint from panel **C**). Representative of >20 experiments. **E** Schematic diagram of the genome-wide synthetic lethality screen to identify genes that are synthetic-lethal with NF2 loss-of-function. **F** Scatter plot of the abundance of each sgRNA from the library at passage 7 (endpoint) normalized to passage 0 for both NF2-WT (x-axis) and NF2-KO#1 (y-axis) cells. Each dot represents a single sgRNA, and abundance is calculated as the log2 Fold Change (FC) of passage 7 / passage 0. **G** Top hits showing a differential proliferative/viability effect in NF2 KO cells compared to NF2-WT cells. Source data are provided as a Source Data file.

protein, and display the expected decrease in YAP phosphorylation and increase in Akt phosphorylation (Fig. 1A, Suppl. Fig. 1B). As expected from the fact that Hippo/YAP signaling mediates contact inhibition[16], the NF2-KO lines have a similar proliferation rate as NF2-WT cells at low cell densities (Fig. 1B), but then start diverging at higher confluence and fail to arrest proliferation at a cell density that causes contact inhibition in NF2-WT cells (Fig. 1C). Instead, at high densities, NF2-KO cells pack very tightly, lose their elongated morphology, and start growing on top of each other (Fig. 1D). Compared to NF2-WT cells, NF2-KO cells acquire characteristics of transformed cells such as the ability to form anchorage-independent colonies in a soft-agar assay (Suppl. Fig. 1C). A comparison of gene expression in NF2-KO vs NF2-WT cells by RNA-seq followed by Gene Ontology enrichment analysis using ShinyGo[17] identified changes in gene sets related to oncogenic signaling (TGF-beta, Hippo, PI3K-Akt) and cell adhesion (Suppl. Fig. 1D).

### Genome-wide sgRNA screen identifies G6PD and ACSL3 as synthetic-lethal partners for NF2

We next introduced Cas9 into the NF2-WT and NF2-KO#1 cells (Suppl. Fig. 1E) and performed a genome-wide CRISPR/Cas9 screen to identify synthetic-lethal partners for *NF2* in Schwann cells (Fig. 1E). The screen was performed using the Toronto CRISPR Human Knockout Library (TKO v3)[18] with 89916 sgRNAs targeting 17232 protein-coding genes. We measured the abundance of each sgRNA after 7 cell passages and compared it to the initial sgRNA representation in the library at passage 0, for both NF2-WT and NF2-KO cells (Fig. 1F). This identified many genes required for cell viability in both cell lines ("essential genes", Fig. 1F). At the other end of the spectrum, the sgRNAs that caused the largest increase in proliferation were all 6 of the sgRNAs targeting *NF2* (green dots, Fig. 1F). This indicates that Schwann cell proliferation is exquisitely sensitive to *NF2*, possibly explaining why loss of NF2 in people leads predominantly to schwannomas[19]. As expected, the sgRNAs targeting *NF2* had little-to-no effect on the proliferation of NF2-KO cells, since these cells already lack NF2 and overproliferate. Of note, during the sgRNA screen, we split cells soon after reaching confluence, hence sgRNAs that either increase the proliferation rate during exponential growth, or blunt contact inhibition, were enriched by the end of the screen. Interestingly, almost all sgRNAs that caused increased proliferation of WT cells had a blunted effect in NF2-KO cells (Fig. 1F), indicating that NF2 signaling is the predominant proliferation-suppressive pathway in Schwann cells.

We asked if there are any target genes whose knockout reduces the number of NF2-KO cells but not NF2-WT cells. Figure 1G shows the top genes sorted by their differential effect on NF2-KO versus NF2-WT cells. We excluded from further consideration any gene that causes increased proliferation of NF2-WT cells, since these are potentially tumor suppressors and could lead to tumors if targeted pharmacologically in patients. Likewise, we excluded any gene that causes reduced cell numbers in NF2-WT cells since targeting them pharmacologically could be toxic. From this, we identified *G6PD* as a top hit, as it causes no proliferative defects in NF2-WT cells but reduces the number of NF2-KO cells, and *ACSL3* as a second hit that has a mild negative effect on NF2-WT cells. G6PD in particular caught our attention because 400 million people worldwide are deficient for G6PD, and people with reductions in G6PD activity down to ~10% of normal have little or no phenotypes as long as oxidative triggers are avoided by lifestyle management[20]. This suggests G6PD could potentially be targeted pharmacologically with an acceptable side-effect profile. Likewise, *ACSL3* knockout mice are born at the expected Mendelian ratio with no obvious defects during development or adulthood[21,22].

### Inhibition of G6PD causes death of NF2-KO Schwann cells and impairs growth of NF2-KO xenograft tumors

We next confirmed that G6PD loss-of-function causes synthetic lethality in combination with NF2 loss-of-function using 4 different targeting modalities – sgRNAs, shRNAs, siRNAs, and a small molecule inhibitor. We performed growth and viability curves with either NF2-WT or NF2-KO cells transduced with sgRNAs targeting either *G6PD* or a negative control locus (*AAVS1*) (Fig. 2A-C): NF2-WT and NF2-KO cells both showed a similar increase in cell number over the course of 4 days when transduced with the negative-control sgRNA (Fig. 2A). Although *G6PD* loss-of-function (Fig. 2C) showed no significant effect on NF2-WT cell numbers, it reduced the number of NF2-KO cells by roughly 50% at day 4 (Fig. 2A). The reduction in cell number was due at least in part to a significant, 6-fold increase in cell death upon *G6PD* loss-of-function in NF2-KO cells, assessed by CellTox (Fig. 2B). Next, we knocked-down *G6PD* using an siRNA that targets a different sequence than the sgRNA. siRNA-mediated knockdown of *G6PD* also reduced the number of NF2-KO but not NF2-WT Schwann cells (Suppl. Fig. 2A-C). A small-molecule inhibitor for G6PD (G6PDi-1) was recently reported[23]. Four days of treatment with this inhibitor reduced the number of NF2-KO cells but not NF2-WT cells (Fig. 2D) with a corresponding increase in cell death in NF2-KO cells (Suppl. Fig. 2D). This experimental setup cannot yield large differences in cell numbers because after 4 days the wildtype cells become confluent in the dish and stop proliferating. If instead the cells are split every 2 days and maintained either in the presence or absence of G6PDi-1 for 3 passages, this reveals a very large drop in the viability of NF2-KO cells treated with G6PDi-1 (Fig. 2E-F). The synthetic lethality was also observed if we grew the cells in a medium containing 5% FBS instead of 10% FBS, and hence in conditions of lower growth factor stimulation (Suppl. Fig. 2E).

We next established a schwannoma xenograft model involving subcutaneous injection of NF2-KO Schwann cells into NOD SCID gamma (NSG) mice. In agreement with our soft-agar assays (Suppl. Fig. 1C), NF2-WT Schwann cells do not form subcutaneous tumors, despite using Matrigel, precluding us from testing *G6PD* knockdown in NF2-WT cells in vivo, whereas NF2-KO cells form tumors that grow flat and nodular with necrotic centers (Fig. 2G). For xenograft experiments, we generated NF2-KO lines stably carrying doxycycline (dox)-inducible shRNA constructs targeting either *RLuc* as a negative non-targeting control or two different regions of *G6PD* (Fig. 2H). Again, the knockdown of *G6PD* reduces the proliferation of NF2-KO cells in cell culture (Fig. 2I). Control NF2-KO Schwann cells reliably generated palpable tumors that grew, both in the presence and in the absence of doxycycline (dox) (Suppl. Fig. 2F, Fig. 2J). In contrast, NF2-KO Schwann

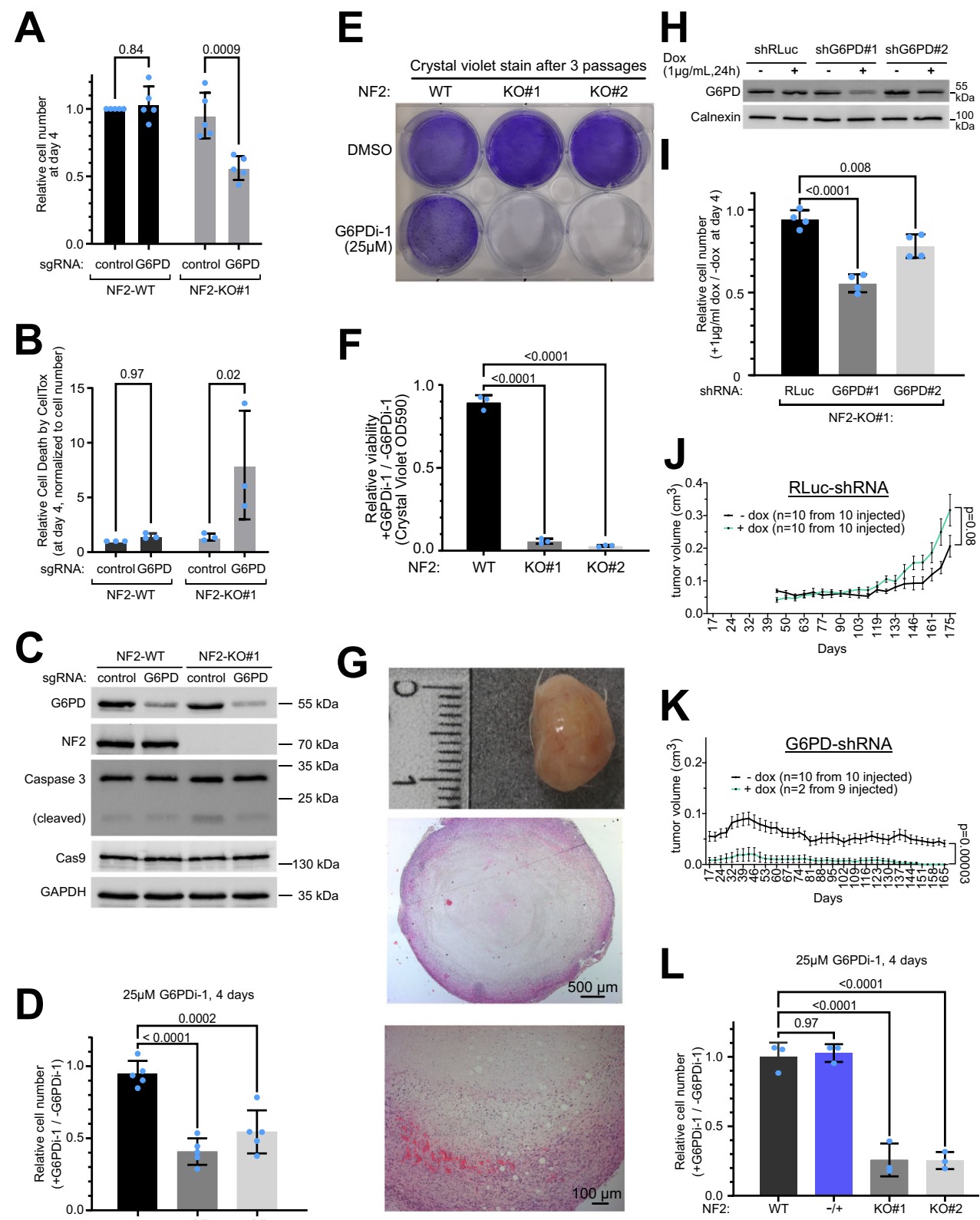

cells with a *G6PD* knockdown (+ dox) yielded much fewer palpable tumors (Suppl. Fig. 2F), and the ones that formed subsequently completely regressed so that no detectable *G6PD* knockdown tumors were present at the end of the follow-up (Fig. 2K). Although tumors containing *G6PD* shRNA in the absence of dox reliably formed palpable tumors that grew (Suppl. Fig. 2F, G), they grew less well than control tumors, likely due to leakiness of the inducible shRNA (Suppl. Fig. 2H).

Nonetheless, the difference in tumor growth between +dox and -dox was highly significant ($p = 0.00003$).

## A 50% inhibition of G6PD is sufficient for synthetic lethality

An important issue for the possible future clinical application of these findings is to understand the degree of G6PD inhibition needed to cause the death of NF2-KO Schwann cells. 400 million people

**Fig. 2 | Validation of NF2/G6PD synthetic lethality. A–C** sgRNA-mediated depletion of G6PD reduces the viability of NF2-KO but not NF2-WT cells. **A** Relative cell number 4 days after seeding (7 days postinfection), assessed by CellTiter-Glo. **B** Cell toxicity analyzed with CellTox, normalized to cell number. **C** Immunoblot to assess G6PD knock-out efficiency. Bars = mean ± SD. Significance by two-way ANOVA and Sidak's multiple comparisons test. $n = 5$ (**A**) or 3 (**B**) independent experiments in biological quadruplicates. **D** Pharmacological inhibition of G6PD reduces the viability of NF2-KO but not NF2-WT Schwann cells. Bars = mean ± SD. Significance by one-way ANOVA and Dunnett's multiple comparisons test. $n = 5$ biological replicates. **E–F** Pharmacological inhibition of G6PD strongly reduces viability of NF2-KO cells, as seen if the cells are passaged 3 times in the presence or absence of G6PDi-1. Passaging prevents wild-type cells from reaching confluence and no longer proliferating. Cells stained with crystal violet. **E** Representative image. **F** Significance by one-way ANOVA and Dunnett's multiple comparisons test. $n = 3$ biological replicates. Bars = mean ± SD. **G** NF2-KO Schwann cells form tumors when implanted subcutaneously in NOD SCID gamma mice. Top panel: the macroscopic image of the excised tumor. Middle & bottom panels: hematoxylin and eosin stained tumor at two different magnifications. Representative of 10 tumors. **H–I** G6PD knockdown with shRNAs impairs the viability of NF2 KO cells. **H** Immunoblot control of G6PD knockdown. **I** Relative cell number 4 days after induction of control or G6PD shRNA, each compared to non-induced controls (DMSO). Bars = mean ± SD. Significance by one-way ANOVA and Dunnett's multiple comparisons test. $n = 4$ biological replicates. **J–K** G6PD knockdown reduces the growth of NF2-KO Schwann cell tumor xenografts. Tumor volumes as a function of time of NSG mice injected subcutaneously and treated +/- Dox for 25 weeks. Renilla Luciferase (RLuc) shRNA = negative control. Animals were randomized into groups. Graphs show mean volumes; error bars=SEM. Significance by multiple unpaired t-test comparison analysis. **L** Inhibition of G6PD does not kill NF2-/+ heterozygous Schwann cells. Significance by one-way ANOVA and Dunnett's multiple comparisons test. $n = 3$ biological replicates. Bars = mean ± SD Source data are provided as a Source Data file.

worldwide are deficient for G6PD to varying degrees[20]. From this, we know that inhibition of G6PD below 5-10% of residual activity (Class I) leads to jaundice and chronic haemolytic anemia, whereas people with higher activity levels have few or no clinical symptoms[20]. To study this in more detail, we used pharmacological inhibition of G6PD, since G6PD activity can be titrated more precisely pharmacologically than with shRNA-mediated knockdown. Our previous results indicated that 25 μM G6PDi-1 is sufficient to cause the death of NF2-KO cells (Fig. 2D). We therefore performed a titration of G6PDi-1 and measured G6PD activity, and found that 25 μM G6PDi-1 leads to roughly 50% inhibition of G6PD (Suppl. Fig. 2I). Hence this is within the range of G6PD inhibition that would be clinically tolerable.

### G6PD inhibition does not kill NF2 +/- heterozygous cells

Often NFII patients are heterozygous for NF2 loss-of-function in many of the cells in their bodies, with loss or mutation of the remaining *NF2* allele leading to tumor formation in some cells. Hence another important consideration is whether G6PD inhibition kills NF2 heterozygous cells, because this would lead to severe toxicity. To test this, we again CRISPRized the ipn02.3 2λ wildtype Schwann cell line with the same sgRNAs that we used to generate the NF2-KO cells, except this time, we screened for heterozygous loss of *NF2* rather than homozygous loss. This yielded a Schwann cell line that is isogenic to ipn02.3 2λ except it has a frame-shift mutation and a premature stop codon on one *NF2* allele (Suppl. Fig. 2J). The second allele has the loss of a triplet, leaving the rest of the coding sequence in-frame, from which protein is produced (Suppl. Fig. 2K) that is predicted to lack one amino acid. This line ("NF2-/+") therefore has a level of NF2 function that is equal to, or less than, that of heterozygous cells. NF2-/+ cells proliferate like NF2-WT cells (Suppl. Fig. 2L), in agreement with previous studies showing that even a small amount of NF2 protein is sufficient to provide full NF2 activity[24,25]. Interestingly, pharmacological inhibition of G6PD did not lead to lethality of the NF2-/+ cells (Fig. 2L).

### Inhibition of ACSL3 causes death of NF2-KO Schwann cells and impairs growth of NF2-KO xenograft tumors

Next, we confirmed that *ACSL3* knockdown is also synthetic-lethal with NF2 loss-of-function in Schwann cells using different targeting modalities. Knockout or knockdown of *ACSL3* using sgRNAs or siRNAs, respectively, impairs the viability of NF2-KO cells but not NF2-WT cells (Fig. 3A-C, Suppl. Fig. 2A, C). Partial knockdown of *ACSL3* via doxycycline-inducible shRNA constructs impairs proliferation of NF2-KO cells in cell culture (Fig. 3D-E) and also significantly impairs the growth of NF2-KO schwannoma xenografts (Fig. 3F).

### G6PD and ACSL3 are synthetic lethal with NF2 due to oxidative stress

Interestingly, both G6PD and ACSL3 are involved in lipid biogenesis and in fighting oxidative stress. G6PD is the first and rate-limiting enzyme in the pentose phosphate pathway and is the predominant source of NADPH used by cells as a reducing agent to synthesize lipids and to fight oxidative stress[26]. ACSL3 is a member of the acyl-CoA synthetase family which conjugates mono-unsaturated fatty acids to coenzyme-A (CoA) for lipid biogenesis, thereby reducing the susceptibility of plasma membrane lipids to oxidation[27]. We therefore asked whether *ACSL3* and *G6PD* are synthetic lethal with *NF2* due to oxidative stress and/or lipid biogenesis.

Since NF2-KO cells proliferate more than NF2-WT cells, we hypothesized they might either have a higher requirement for lipid biogenesis, or a higher requirement for reducing equivalents, which are needed for lipid biogenesis. Hence we tested if we could rescue the synthetic lethality of *ACSL3* loss-of-function by supplementing cells either with additional exogenous lipids, or with the antioxidant N-acetylcysteine (NAC). Interestingly, the lipid mix, but not NAC, rescued the synthetic lethality of *ACSL3* (Fig. 4A-B), suggesting the synthetic lethality is related to the lipid conjugation function of ACSL3. ACSL3 conjugates mono-unsaturated fatty acids (MUFAs) onto CoA for lipid biosynthesis, reducing the abundance of polyunsaturated fatty acids (PUFAs) in cell membranes, which are prone to peroxidation and hence to induction of ferroptosis[27]. Indeed, treatment of cells with the ferroptosis inhibitor Liproxstatin rescued the lethality caused by *ACSL3* knockout in NF2-KO cells (Fig. 4C), indicating the cells are dying by ferroptosis. An analysis of lipid peroxidation levels with Bodipy C11 revealed that NF2-KO cells have mildly but significantly elevated levels of lipid peroxidation compared to NF2-WT cells (Fig. 4D). In sum, this suggests NF2-KO cells are poised to undergo ferroptosis compared to WT cells due to elevated lipid oxidation. This is conceptually consistent with a previous report that NF2 mutant mesothelioma cells are sensitive to ferroptosis[28].

Unlike for ACSL3, the synthetic lethality between *G6PD* and *NF2* cannot be rescued by liproxstatin (Fig. 4E) or by supplementing cells with a lipid mix (Fig. 4F). Nonetheless, the synthetic lethality with *G6PD* is rescued by addition of the antioxidant NAC (Fig. 4G-H), suggesting again that the underlying cause of the *NF2/G6PD* synthetic lethality is oxidative stress.

The form of cell death induced by combined loss of *NF2* and *G6PD* is not clear. Since it is not rescued by Liproxstatin (Fig. 4E) it is not ferroptosis. Although we see some caspase cleavage when we knock out *NF2* and *G6PD* with sgRNAs (Fig. 2C), this is likely due to the viral infection since it is also present in the control cells treated with control sgRNA (lane 1). Indeed, pharmacological inhibition of *G6PD* in NF2-KO cells does not cause caspase cleavage (Suppl. Fig. 3A). The lethality is also not rescued by Emricasan, a pan-caspase inhibitor (Suppl. Fig. 3B)

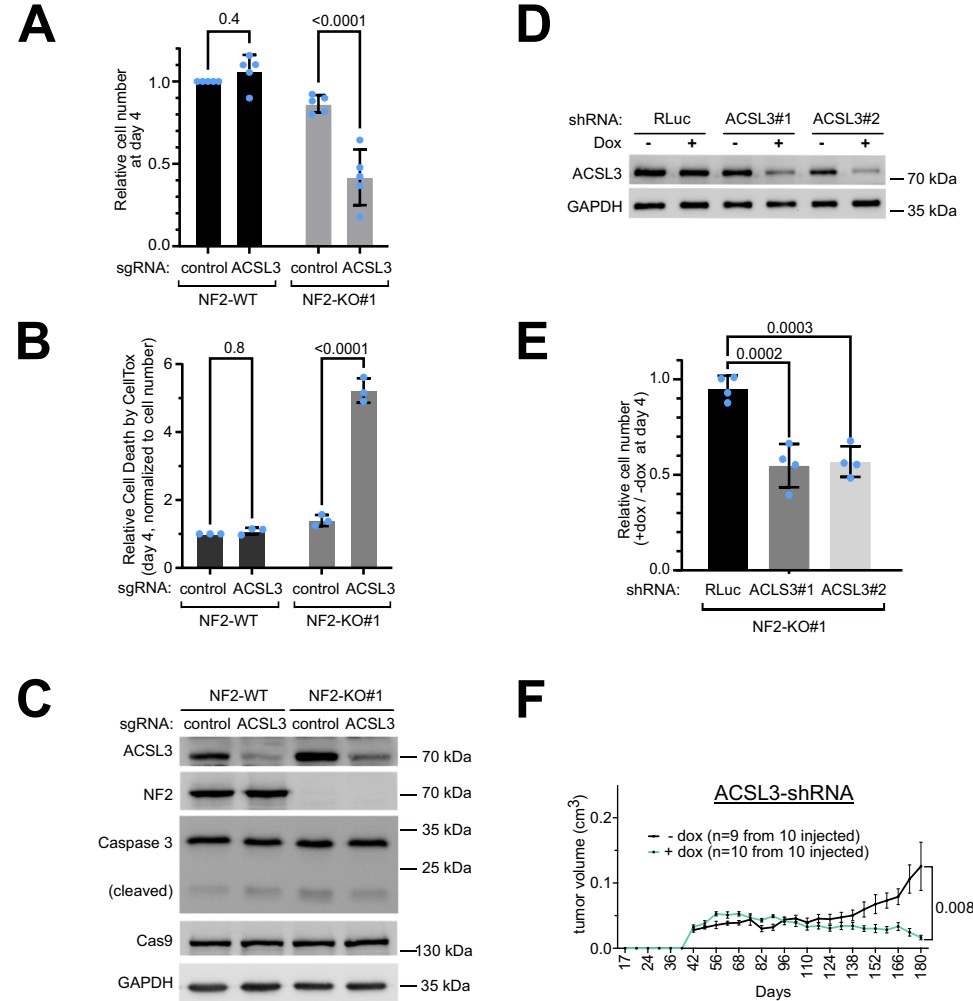

**Fig. 3 | Validation of ACSL3/G6PD synthetic lethality. A–C** sgRNA-mediated depletion of ACSL3 causes reduced viability of NF2-KO cells but not NF2-WT cells. (A) Relative cell number 4 days after seeding (7 days post infection), assessed by CellTiter-Glo. Bars = mean ± SD. (**B**) Cell toxicity analyzed with CellTox. Values are normalized to total cell number (CellTiter-Glo). Bars = mean ± SD. **C** Representative control immunoblots to assess ACSL3 knock-out efficiency. **A–B** Significance by two-way ANOVA and Sidak's multiple comparisons test. *n* = 5 (**A**) or 3 (**B**) biological replicates. **D–E** Induction of ACSL3 knockdown with two different shRNAs impairs viability of NF2 KO cells. **D** Immunoblot control of ACSL3 knockdown upon Dox treatment (1 μg/mL for 24 hrs) of human Schwann cell lines stably transfected to carry indicated shRNAs. **E** Relative cell number 4 days after induction (1 μg/ml doxycycline) of control or ACSL3 shRNA, each compared to non-induced controls (DMSO). Bars = mean ± SD. Significance by one-way ANOVA and Dunnett's multiple comparisons test. *n* = 4 biological replicates. **F** ACSL3 knockdown reduces growth of NF2-KO Schwann cell tumor xenografts. Tumor volume as a function of time of NSG mice injected subcutaneously with indicated Schwann cell lines and treated +/-Dox for 25 weeks. Animals were randomized into the +dox vs -dox groups. Graphs show mean volumes; error bars=SEM. Significance by multiple unpaired t-test comparison analysis. Source data are provided as a Source Data file.

suggesting they are not dying via caspase-dependent apoptosis, but some other cell death mechanism. As a control, we verified that the same concentration of Emricasan efficiently blocks caspase cleavage induced by staurosporin (Suppl. Fig. 3C).

## NF2-KO Schwann cells have a more oxidized redox state

We noticed in our RNA-seq data that NF2-KO cells have lower expression of several enzymes that fight oxidation. For instance, expression of all three members of the aldosterone reductase family 1 (AKR1C1, 2 and 3) which reduce lipid peroxides to lipid alcohols[29] thereby protecting cells from ferroptosis[30], are lower in two independent NF2-KO lines compared to control cells (Fig. 5A). This is also visible at the protein level for the one AKR1 member we tested, AKR1C3 (Fig. 5B). In addition, the level of GPX4, a glutathione-dependent lipid peroxidase that plays a key role in protecting cells from ferroptosis[30], is also mildly reduced (Fig. 5B). The attenuated levels of AKR1C1, AKR1C2, AKR1C3 and GPX4 could explain why NF2-KO cells are sensitized to ferroptosis when *ACLS3* is inhibited[30].

Amongst the most down-regulated genes in NF2-KO Schwann cells is *Malic Enzyme 1* (*ME1*) (Fig. 5C). ME1 and G6PD are two of the four enzymes that generate cytosolic NADPH, which is used by cells to fight oxidative stress, with G6PD being the predominant NADPH source and ME1 the second main source[31]. By immunoblotting, we found that NF2-KO cells not only have strongly reduced levels of ME1 (Fig. 5D), but also mildly reduced levels of G6PD. Consistent with this, NF2-KO cells have reduced ME1 and G6PD activity (Fig. 5E-F) and a lower ratio of reduced NADPH to oxidized NADP+ (Fig. 5G). In sum, NF2-KO Schwann cells have lower levels of reducing equivalents (NADPH) and lower levels of enzymes that help counteract oxidative stress.

We next asked if these effects are a general feature of loss of NF2 in Schwann cells. First, we knocked-down *NF2* using two independent siRNAs in primary human Schwann cells and found that this leads to a drop in ME1, AKR1C3 and GPX4 also in these cells (Suppl. Fig. 4A). Of note, these primary cells have an independent genetic background from the immortalized ipn02.3 2λ Schwann cell line that we used above. Next, we analyzed the HEI-193 human schwannoma line, which

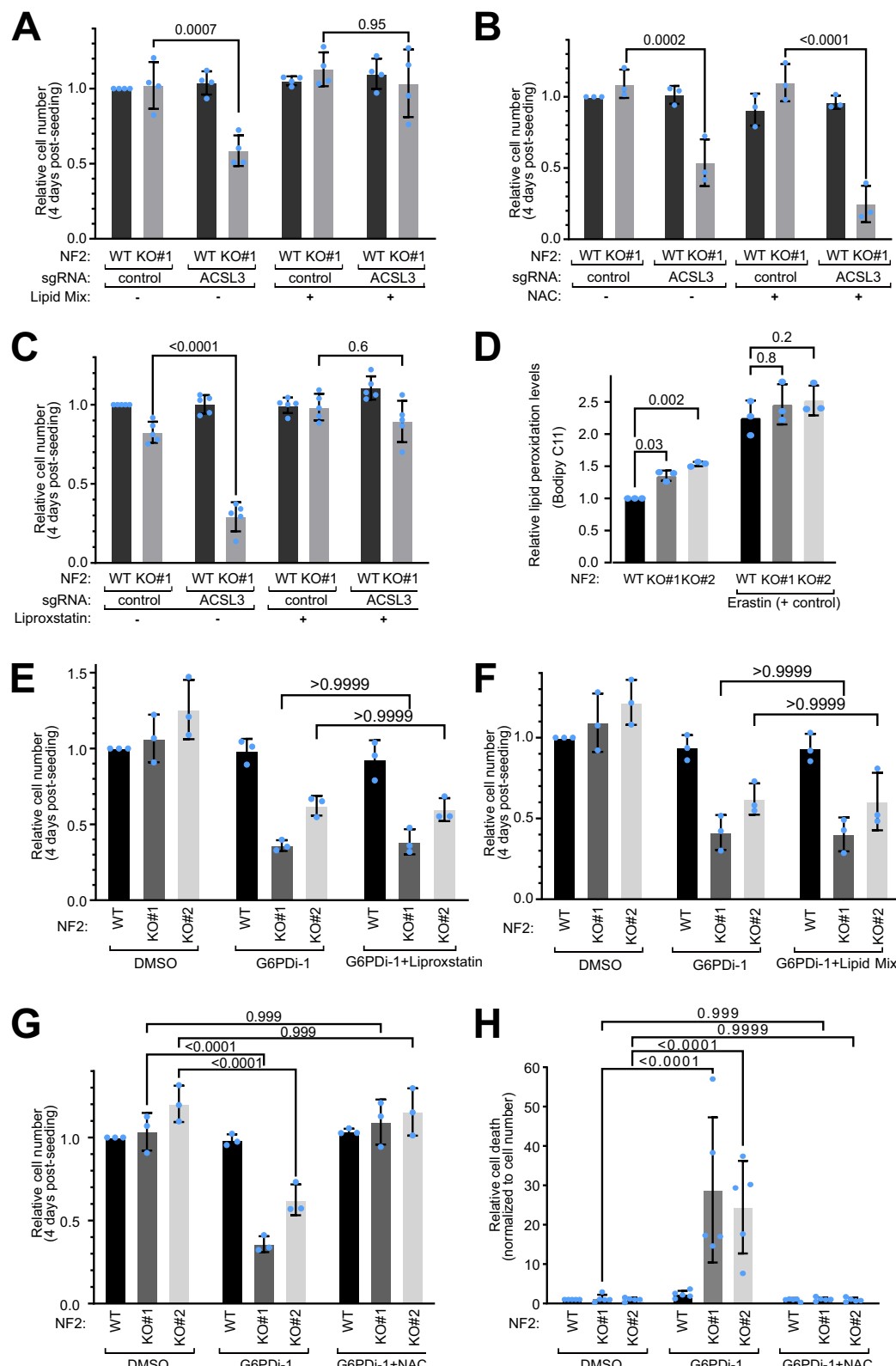

has a point mutation that causes a splicing defect in the NF2 transcript, and thereby a partial NF2 loss-of-function[32]. As expected, these cells express NF2 'isoform 3' which runs lower than full-length NF2, and they have reduced YAP phosphorylation (Suppl. Fig. 4B). In addition, they have low levels of ME1 and AKR1C3 (Suppl. Fig. 4B). Interestingly, HEI-193 cells have elevated levels of malic enzyme 3 (ME3), which also synthesizes NADPH, perhaps as a compensatory effect.

Next, we tested whether reduced levels of AKR1, 2, 3, and ME1 can also be observed in primary schwannomas from patients. To this end, we reanalyzed data from Gugel et al.[33], which profiled non-irradiated vestibular schwannomas (VS) from 49 patients—36 sporadic VS, 13 NF2 mutant VS, and 9 cystic VS—and compared them to 7 control vestibular nerve samples. This revealed that vestibular schwannomas have significantly reduced expression of all four genes (Fig. 5H-I) as well as

**Fig. 4 | Characterization of the synthetic lethality caused by G6PD or ACSL3.**
**A–B** The synthetic lethality between ACSL3 and NF2 is rescued by treating cells with a lipid mix (1%) (A) but not by the antioxidant n-acetylcysteine (NAC, 1 mM) (**B**). Relative cell number (CellTiter-Glo) normalized to the control sgRNA for each condition. Bars = mean $\pm$ SD. Significance by two-way ANOVA and Tukey's multiple comparisons test. $n = 4$ (**A**) or 3 (**B**) biological replicates. **C** Synthetic lethality of NF2 and ACSL3 is rescued by Liproxstatin. Relative cell number (CellTiter-Glo) upon treatment with the ferroptosis inhibitor Liproxstatin. Values are normalized to control sgRNA. Bars = mean $\pm$ SD. Statistical analysis shown is a two-way ANOVA and Tukey's multiple comparisons test; $n = 5$ biological replicates. **D** NF2 knockout Schwann cells have mildly elevated lipid peroxidation levels, assessed using BOD-IPY 581/591 C11 and flow cytometry. Ratio of oxidized/reduced lipids were normalized to untreated NF2-WT background levels. Erastin treatment (10μM-5hrs)

was used as a positive control for lipid peroxidation. Bars = mean $\pm$ SD. Significance by two-way ANOVA and Tukey's multiple comparisons test. $n = 3$ biological replicates. **E–G** The synthetic lethality between NF2 and G6PD is rescued by (**G**) the antioxidant n-acetylcysteine (NAC, 1 mM), but not by (**E**) Liproxstatin (1 μM), or by (**F**) supplementation with 1% Lipid Mix. Relative cell numbers, normalized to the DMSO control treatment condition, assayed by CellTiter-Glo. G6PDi-1 used at 25 μM. Bars = means $\pm$ SD. Significance by two-way ANOVA and Tukey's multiple comparisons test. $n = 3$ biological replicates. **H** G6PDi-1 induces cell death (assayed by CellTox) in NF2-KO Schwann cells, and is rescued by NAC. Cell death values are normalized to cell number (CellTiter-Glo). Bars = mean $\pm$ SD. Significance by two-way ANOVA and Tukey's multiple comparisons test. $n = 5$ independent experiments each with biological quadruplicates. Source data are provided as a Source Data file.

reduced expression of *G6PD* (Suppl. Fig. 4C). Of note, these measurements were done by microarray rather than RNA-seq, and the tumor samples contain additional cell types besides Schwann cells, both of which reduce the magnitude of the gene expression changes that can be observed. The fact that all four genes drop in expression in NF2 mutant vestibular schwannomas in patients, as they do upon loss of NF2 in the sural-nerve derived ipn02.3 2λ Schwann cell line that we use for most of our experiments here, suggests that this biology is conserved in Schwann cells from different locations of the body. In sum, loss of NF2 in Schwann cells is associated with reduced expression of several anti-oxidative genes in multiple different genetic backgrounds across different cell lines and patients.

### Low ME1 levels can cause synthetic lethality with G6PD
The data presented above raise the mechanistic hypothesis that inhibition of G6PD is lethal in NF2-KO cells because they have a more oxidized redox state. This oxidized redox state is due in part to reduced expression of NADPH-producing enzymes such as ME1 (Suppl. Fig. 5A), although we cannot exclude that loss of NF2 might also lead to increased oxidative stress. According to this hypothesis, NF2-KO cells can buffer these redox changes by relying on NADPH produced by G6PD, but if G6PD is also inhibited this leads to cell death.

To test this hypothesis, we first assessed the redox status of cells via the ratio of oxidized to reduced glutathione. Although NF2-KO cells have a lower ratio of reduced NADPH to oxidized NADP+ (Fig. 5G), this imbalance does not translate into a change in the basal oxidation state of glutathione in NF2-KO cells compared to NF2-WT cells (Fig. 6A, bars 4 & 7 versus 1) nor in elevated ROS levels (Suppl. Fig. 5B), suggesting that the remaining NADPH is sufficient for NF2-KO cells to maintain a proper redox balance further downstream. Likewise, inhibition of G6PD in wildtype cells reduces the NADPH/NADP+ ratio (Suppl. Fig. 5C) but does not cause significant oxidation of glutathione (bar 2 vs 1, Fig. 6A). In contrast, G6PD inhibition in NF2-KO cells causes the glutathione pool to become significantly more oxidized (bars 5 & 8 versus 4 & 7 respectively, Fig. 6A). This is rescued by addition of NAC (Fig. 6A), in agreement with the rescue of lethality by NAC (Fig. 4G). Thus NF2-KO cells rely more strongly on G6PD for production of reducing equivalents compared to wild-type cells.

We next asked whether inhibition of ME1 in NF2-WT cells recapitulates the phenotype, causing them to become sensitive to G6PD inhibition. To this end, we first performed a titration of an ME1 inhibitor (Suppl. Fig. 5D) and found that 10 μM ME1-i causes a similar drop in ME1 activity as we observe in NF2-KO cells (Fig. 5E). Consistent with our mechanistic hypothesis, pharmacological inhibition of only G6PD did not reduce the viability of NF2-WT cells, nor did inhibition of ME1 only (Fig. 6B). Combined inhibition of G6PD and ME1, however, caused a significant, synergistic drop in the viability of NF2-WT cells (Fig. 6B), indicating that the two enzymes function redundantly in Schwann cells to support viability, and that their combined inhibition is sufficient to explain why G6PD inhibition causes NF2-KO Schwann cells to die.

We also tested the reverse - whether re-expression of ME1 in NF2-KO cells rescues their viability upon G6PD inhibition. To this end we generated a monoclonal NF2-KO line that re-expresses ME1 (Fig. 6C). Re-expression of ME1 was sufficient to rescue the death of NF2-KO cells upon G6PD inhibition (Fig. 6D).

### Mechanism of ME1 transcriptional regulation
We do not know, mechanistically, why *ME1* mRNA levels are reduced upon loss of *NF2* in Schwann cells. One pathway that is activated in the NF2-KO cells is the PI3K/Akt pathway (Fig. 1A). Pharmacological inhibition of PI3K with LY294002 for 24 hrs, however, does not restore ME1 levels to normal (Suppl. Fig. 6A) indicating that ME1 is not reduced due to elevated PI3K activity. Another consequence of loss of NF2 is the activation of YAP/TAZ, which in turn interact with several transcription factors to regulate gene expression: TEAD transcription factors, AP-1 transcription factors, CDK7, p73, and RUNX1/2[34]. We pharmacologically inhibited the TEADs with VT-104[35] and used ANKR1 as a canonical YAP/TAZ target to readout activity. As expected, NF2-KO cells have *ANKR1* mRNA levels that are 1000x higher than in NF2-WT cells (Suppl. Fig. 6B), and *ME1* levels that are low (Suppl. Fig. 6C). Inhibition of the TEADs, however, does not bring *ANKR1* levels back down to the level of NF2-WT cells. The levels of *ANKR1* in NF2-KO cells + VT-104 are still 500-fold higher than in NF2-WT cells (Suppl. Fig. 6B) suggesting that YAP/TAZ rely mainly on other partners to drive gene expression in Schwann cells. Correspondingly, inhibition of the TEADs also did not restore *ME1* levels (Suppl. Fig. 6C). Thus, further work will be required to understand how YAP/TAZ regulate gene expression in Schwann cells and how the loss of NF2 leads to low *ME1* expression. Interestingly, we noticed that *ME1* levels are not only reduced in NF2 mutant schwannomas from patients (Fig. 5H) but also in NF2 mutant meningiomas from patients (Suppl. Fig. 6D, E).

### HEI-193 cells become sensitized to G6PD inhibition upon loss of NF2
We tested whether the concept that Schwann cells become sensitized to oxidative stress upon loss of NF2 also holds true in other Schwann cells. Unfortunately, very few immortalized NF2-wildtype human Schwann cell lines exist. Primary cells are not suitable because they do not remain proliferative for a sufficient number of doublings to perform a genetic manipulation followed by proliferation curves. We therefore turned to HEI-193 cells which express NF2, albeit a mutant version (Suppl. Fig. 4B). We generated NF2-KO HEI-193 cells, and found that they have less phospho-YAP compared to the parental HEI-193 line (Suppl. Fig. 7A), indicating that the NF2 in HEI-193 cells is still partially active. Just as we observed in the ipn02.3 2λ line, knockout of NF2 in HEI-193 cells sensitized them to G6PD inhibition, and this is rescued by NAC, indicating that they die due to oxidative stress (Suppl. Fig. 7B). Just as in the ipn02.3 2λ line, knockout of NF2 alone, or inhibition of G6PD alone, did not cause an increase in glutathione oxidation, but combined inhibition of G6PD and NF2 did (Suppl. Fig. 7C). In

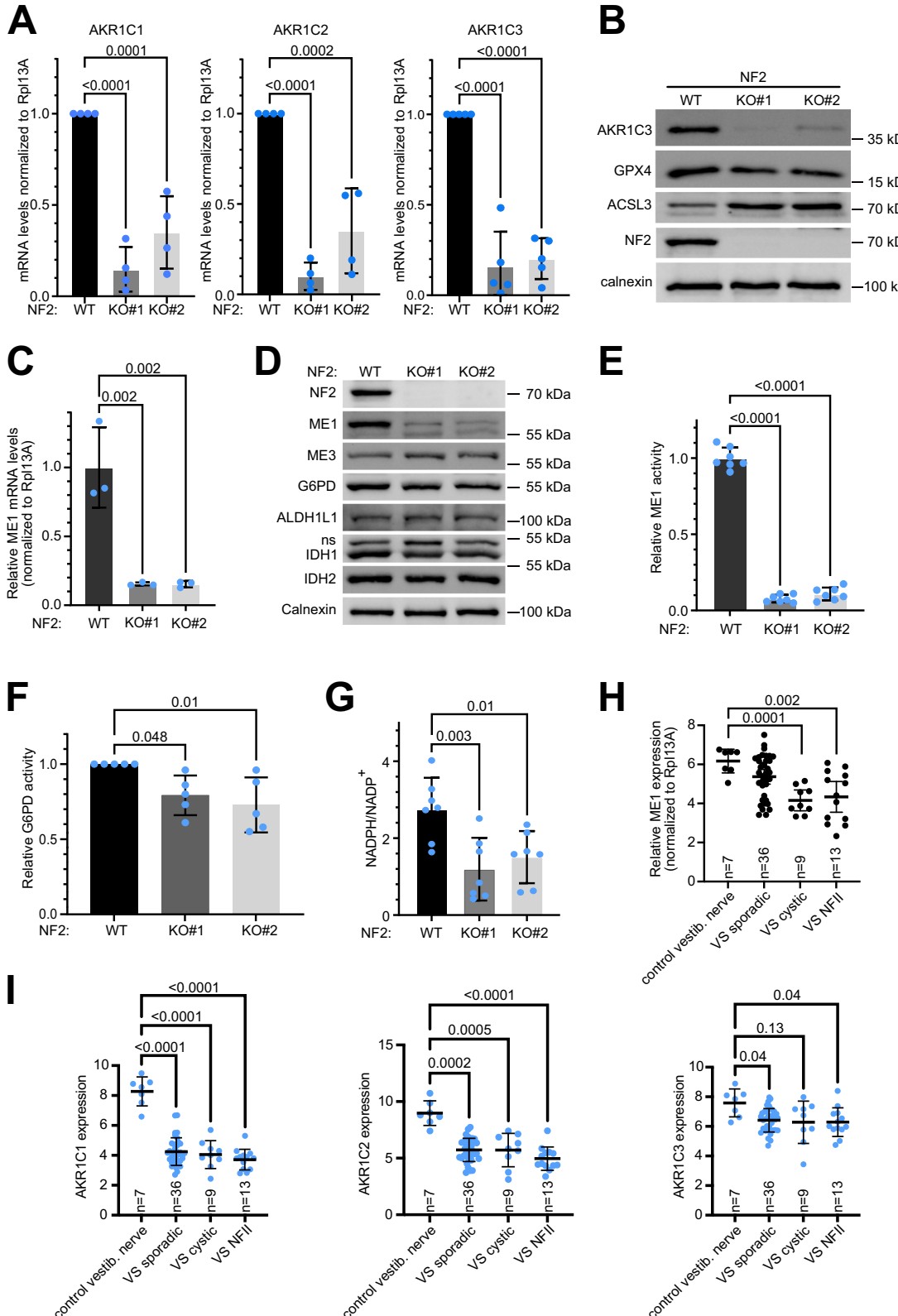

conclusion, also HEI-193 cells become sensitized to G6PD inhibition when NF2 is knocked out. We noted, however, that the parental HEI-193 line already has low ME1 levels (Suppl. Fig. 4B), which do not drop further upon complete NF2 knockout (Suppl. Fig. 6A). Thus, ME1 expression is not the sole reason why the loss of NF2 causes cells to become more oxidized, in agreement with the many changes in redox genes caused by NF2 loss (Fig. 5).

## Synthetic lethality between NF2 and G6PD is specific to Schwann cells

The synthetic lethality between *NF2* and *G6PD* could be specific to Schwann cells, or it could be a general phenomenon observed in different cell types. To distinguish these two options, we introduced an NF2-KO using CRISPR/Cas9 into a variety of different cancer and non-cancer cell lines. In some cells, such as HeLa cells (cervical

**Fig. 5 | NF2-KO Schwann cells are more oxidized than control cells. A** NF2 KO Schwann cells have reduced AKR1C1, 2 and 3 mRNA levels compared to isogenic NF2-WT controls, assayed by Q-RT-PCR. Bars = mean± SD. Dots = biological replicates. Significance by one-way ANOVA and Dunnett's multiple comparisons test. *n* = 4 biological replicates **B** NF2 KO Schwann cells have reduced AKR1C3 and increased ACSL3 protein levels compare to isogenic NF2-WT controls. **C** NF2-KO Schwann cells have reduced ME1 mRNA levels, assayed by Q-RT-PCR, compared to NF2 WT isogenic controls. Bars = mean ± SD. *n* = 3 biological replicates, Significance by one-way ANOVA and Dunnett's multiple comparisons test. **D** NF2-KO Schwann cells have reduced levels of ME1 protein compared to NF2-WT isogenic controls. **E** NF2-KO Schwann cells have reduced ME1 activity compared to NF2 WT isogenic controls. Bars = mean ± SD. *n* = 7 biological replicates. Significance by two-way ANOVA and Tukey's multiple comparisons test. **F** NF2-KO Schwann cells have reduced G6PD activity compared to NF2 WT isogenic control cells. Bars = mean ±

SD. Significance by one-way ANOVA and Dunnett's multiple comparisons test. *n* = 5 biological replicates. **G** NF2-KO Schwann cells have a reduced NADPH/NADP⁺ ratio compared to isogenic controls. Dots = biological replicates. Bars = mean± SD. Significance by one-way ANOVA and Dunnett's multiple comparisons test. *n* = 8 biological replicates. **H** ME1 mRNA levels are significantly reduced in NF2 mutant and cystic vestibular schwannomas from patients compared to control vestibular nerve. Data are re-analyzed from[33]. Dots = individual patients. Bars = mean ± SD Significance by Brown-Forsythe and Welch ANOVA and Dunnetts's multiple comparisons test. **I** AKR1C1, 2 and 3 mRNA levels are consistently reduced in different types of vestibular schwannomas (VS: sporadic, cystic and NF2) compared to control vestibular nerve. Data are re-analyzed from[33]. Dots = individual patients. Bars = mean± SD. Significance by Brown-Forsythe and Welch ANOVA and Dunnetts's multiple comparisons test. Source data are provided as a Source Data file.

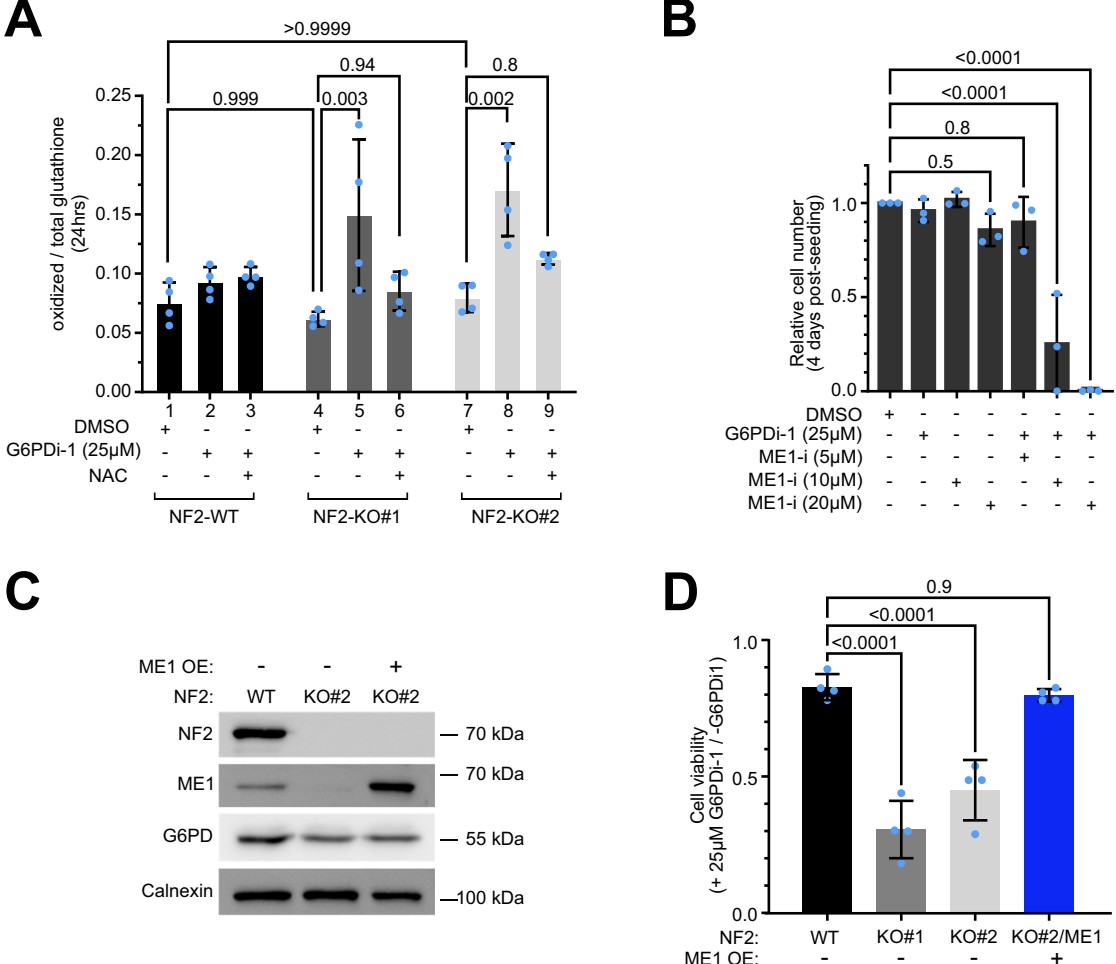

**Fig. 6 | NF2 and ME1 are synthetic lethal in Schwann cells. A** Relative oxidized glutathione levels are increased in NF2-KO Schwann cells after treatment with G6PDi-1 and rescued by n-acetylcysteine (NAC). Bars = mean± SD. Significance by two-way ANOVA and Tukey's multiple comparisons test. *n* = 4 biological replicates. **B** G6PD inhibition and ME1 inhibition display synthetic lethality in Schwann cells. Cell number (by CellTiter-Glo) of NF2-WT cells treated +/- G6PDi-1 +/- ME1 inhibitor, normalized to the non-treated DMSO condition. Bars = mean± SD. Significance by one-way ANOVA

and Dunnett's multiple comparisons test. *n* = 3 biological replicates. **C, D** Re-expression of ME1 rescues the sensitivity of NF2-KO Schwann cells to G6PD inhibition. **C** Immunoblot of ME1 levels in NF2-WT cells, NF2-KO cells, and a monoclonal line of NF2-KO cells transfected to express ME1. **D** Relative cell numbers of the indicated cell lines 4 days after treatment with 25 μM G6PDi-1, normalized to untreated cells. Bars = mean ± SD. Significance by one-way ANOVA and Dunnett's multiple comparisons test. *n* = 4 biological replicates. Source data are provided as a Source Data file.

cancer line) or human umbilical vein endothelial cells (HUVECs), loss of NF2 does not lead to the reduction of YAP phosphorylation, indicating that the NF2 pathway is not active in these cells (Suppl. Fig. 8A, B). In some cells, such as the osteosarcoma U2OS line, loss of NF2 leads to a strong reduction in YAP phosphorylation (Suppl. Fig. 8C), however, these cells express little ME1 so no change in ME1

expression can be observed. Finally, in some cells, such as the colorectal carcinoma HCT116 line or immortalized human fibroblasts, NF2-KO causes a drop in YAP phosphorylation, indicating that the NF2 pathway is functional, but does not cause a drop in ME1 (Suppl. Fig. 8D, E). In all cases, loss of NF2 does not sensitize these cells to pharmacological G6PD inhibition (Suppl. Fig. 8A-E). Thus, the

synthetic lethality between NF2 and G6PD seems to be fairly specific for Schwann cells.

## Discussion

Unlike other tumor types, which usually arise as single events later in life and therefore can be treated with aggressive therapy, NFII tumors are usually first identified in young adolescents and occur recurrently throughout life, therefore necessitating therapeutic approaches with more mild side-effects. We identify here G6PD and ACSL3 as possible therapeutic targets for NFII.

Interestingly, there are some commonalities between the two synthetic-lethal partners that we identified, ACSL3 and G6PD. Both relate to oxidation. In the case of ACSL3, the synthetic lethality involves ferroptosis, which is a process initiated by lipid peroxidation. In the case of G6PD, the synthetic lethality is due at least partly to reduced NADPH production, and it is rescued by the antioxidant NAC. Hence the fact that ACSL3 and G6PD were both identified as top hits in our screen suggests that an underlying shift in redox state is a metabolic vulnerability of NF2 mutant Schwann cells. This is partly due to reduced expression of enzymes used to fight oxidative stress, such as ME1 and the AKR1C family of aldo/keto reductases. However, there may also be increased oxidative stress in NF2-KO cells. For instance, a previous study reported that NF2 mutant cells have elevated lipogenesis[36]. In our RNA-seq data of NF2-KO Schwann cells, we also see mildly elevated expression of some of these genes, such as *FASN*, *SREBF1* and *LPIN1*. This might contribute to an elevated requirement for cytosolic NADPH.

*ACSL3* knockout mice are born at the expected Mendelian ratio with no obvious defects during development or adulthood[21,22]. This suggests ACSL3 may represent a viable target for NFII. Interestingly, ACSL3 has also been proposed as a pharmacological target for mutant KRAS lung cancer, pancreatic cancer, clear cell renal cell carcinoma, and colorectal carcinoma[21,22,37,38]. Thus ACSL3 may be interesting as a possible target for cancer therapy more broadly. One challenge may be that one would need to specifically target ACSL3 and not other members of the ACSL family.

Overall, to us G6PD appears most promising as a drug target because a G6PD inhibitor is likely to have a mild side-effect profile if dosed appropriately, given that 400 million people worldwide are genetically deficient for G6PD[20]. The synthetic lethality between NF2 and G6PD appears to occur at the level of NADPH. Cytosolic NADPH is generated mostly via two enzymes acting redundantly – G6PD and ME1. Our data (Fig. 6A-B), as well as the fact that G6PD deficiency is tolerated in the human population, suggest that as long as one of the two enzymes is well expressed, it generates sufficient NADPH for cells to survive without stress. Inhibition of ME1, however, is sufficient to recapitulate the synthetic lethality with G6PD inhibition (Fig. 6B), indicating that simultaneous loss of both enzymes is detrimental for cell viability. We do not know the transcriptional details why NF2 loss-of-function leads to ME1 inhibition—a subject we are currently investigating.

Interestingly, people with G6PD deficiency are protected from a wide range of cancers, including colorectal cancer (43% risk reduction[39]) gastric cancer (60% risk reduction[40]), and hepatocellular carcinoma (52% risk reduction[40]). Furthermore, G6PD levels are elevated in many tumor entities, such as colorectal cancer[41]. Patients with lower tumor levels of G6PD have significantly better survival from colorectal cancer[41], bladder cancer[42], breast cancer[43], and acute myeloid leukemia[44]. All these data indicate that reduced G6PD has a beneficial effect on cancer incidence and/or progression more broadly.

In sum, we propose here that G6PD may be a promising pharmacological target for NFII.

## Limitations of our study

Although we tried to obtain and test as many different human Schwann cell lines as possible in this study, in the end we were able to test 2 immortalized lines and 1 primary line. In addition, these findings are supported with data from 58 patients. Nonetheless, in the future, it will be important to test additional cell lines. The difficulty is that few NF2-WT human Schwann cell lines exist, and the primary Schwann cells do not expand enough to be able to generate NF2-KO lines by CRISPR/Cas9 to perform proliferation assays.

## Methods

All animal experiments were done in accordance with the guidelines of the responsible national authority, and with approval of the local Governmental Authority for Animal Experimentation (Regierungspräsidium Karlsruhe, Germany, license 35-9185.81/G-30/20).

### Chemical compounds

G6PDi-1 inhibitor was custom synthesized by Otava Chemicals or purchased from Merck/Sigma (#SML2980). Ferroptosis inhibitor Liproxstatin (#SML1414), chemically defined Lipid Mixture (#L0288), InSolution Staurosporin (#569396), Erastin (#7781) and N-Acetyl L-Cysteine/NAC (#A7250) were purchased from Merck/Sigma. Malic Enzyme 1 inhibitor (#HY-124861) was from Hölzel, and Oligomycin (#SAFSO4876) from VWR international. Emricasan (PF 03491390) was purchased from MedChemExpress.

### Cell lines and culturing

The immortalized human Schwann cell line ipn02.3 2λ was a generous gift of Dr. Margaret Wallace (University of Florida). Immortalized human umbilical vein endothelial cells (HUVEC, #INS-CI-1002) and immortalized human fibroblasts (huFIB, #INS-CI-1010) were purchased from InScreenex. HUVEC were cultured on 0,5% gelatin matrix (#INS-SU-10, InScreenex) and expanded in the corresponding complete medium (#INS-ME-1011, InScreenex). HuFIB were grown on collagen coating (#INS-SU-1017) and their respective medium (#INS-ME-1001), both from Inscreenex. Primary human Schwann cells from Creative Bioarrays (#CSC-7715W) were grown on 2 μg/cm² poly-L-lysine (#0403, Sciencell) and cultured with commercial medium (#1701, Sciencell). Parental HEI-193 line was kindly provided by Prof. Valerio Magnaghi (University of Milan). Immortalized Schwann cells (ipn02.3 2λ), HEI-193, Hela, HCT116 and U20S cell lines (ATCC) were cultured in high-glucose Dulbecco's Modified Eagle Medium (#41965-062, Gibco), supplemented with 10% fetal bovine serum (FBS, Sigma) and 100U/ml penicillin/streptomycin (#15140-122, Life Technologies). The absence of mycoplasma was confirmed by regular testing of cell cultures (Eurofins Genomics). No cell identification was performed.

### Generation of NF2-KO and NF2-/+ ipn02.3 2λ cell lines

sgRNA oligos targeting the NF2 gene locus were designed using the ChopChop sgRNA design tool (https://chopchop.cbu.uib.no/). Sequences of oligos used to generate the NF2-KO lines are given in Suppl. Table 2. Oligos were synthesized by Sigma-Aldrich and cloned into the px459 plasmid (#62988, Addgene), according to the published protocol[45]. To generate NF2 knockouts, wild-type cells were transfected with px459 plasmids containing sgNF2 sequences. Schwann cell line ipn02.3 2λ, HUVEC and HuFib lines were transfected using Metafectene Pro reagent (#T040, Biontex), while HCT116, Hela and U2OS were transfected with Lipofectamine 3000 (#L3000001,ThermoFisher Scientific) following the manufacturer's instructions. For the generation of the NF2-/+ line, transfection of NF2-WT cells was done using PEI reagent (#408727, Sigma) at a ratio 1:3. 48 hrs post-transfection, cells were treated with 1.5 μg/ml Puromycin (#P9620, Sigma) for 72 hrs. After Puromycin treatment and expansion of the surviving population, single clone selection was performed via cell dilution.

For the generation of HEI-193 KO lines, subconfluent cells were infected in the presence of 6 μg/ml Polybrene with lentiviral particles harboring Lenti-Cas9-2A-Blast (#73310, Addgene). 48 hrs

postinfection, cells were treated with 5 µg/ml Blasticidin for 72 hrs. A second round of infection was performed in the surviving pooled population, with lentiviruses harboring sgRNA oligo targeting either the control AAVS1 locus or NF2 (exon2) into the lentiviral PLCKO vector (#73311 Addgene). After 3 days of Puromycin selection (1.5 ug/ml), single cell clone expansion was performed.

## Genotyping of NF2KO and NF2-/+ ipn02.3 2λ single clones

The genomic DNA of the selected clones was isolated using DNeasy Blood and Tissue Kit (#69504, Qiagen). For the validation of editing events in single-cell clones, primer pairs were designed 100-150 bp up and downstream of the sgRNA targeting site. Sequences are provided in Suppl. Table 2. The resulting PCR products were purified from an agarose gel using the NucleoSpin Gel and PCR Clean-up kit (#740609, Macherey-Nagel) and introduced into the TOPO™ TA Cloning™ vector (#450640, Invitrogen). After bacterial transformation of the ligation product, TOPO clones were selected (~10 per cell line) for Sanger sequencing, to detect the indels occurring in NF2.

## Generation of ME1 re-expressing NF2-KO Schwann cell line

The ME1 ORF was amplified from a ME1orf Gateway clone (Clone ID #130654902, GPCF, DKFZ, Heidelberg, Germany) using Phusion enzyme (#M0530L, NEB) according to manufacturer's description. The oligos for this PCR are provided in Suppl. Table 2. The ME1 ORF was gel-purified, cloned into PCRII-TOPO (#450640, Thermofischer Scientific) and sequence-verified. The ME1 ORF was then excised from the PCRII-TOPO plasmid using Nhe1 and Not1 restriction enzymes, and ligated into the same sites of a PiggyBAC transposon vector, and verified by restriction mapping. NF2 KO Schwann cell lines were then transfected with this vector together with a PiggyBAC transposase plasmid in a 1:1 ratio using PEI. After puromycin selection, serial dilutions of the surviving population were performed in order to analyze single-cell clones.

## Generation of lentiviruses and infections

Lentiviral particles were produced by transfecting Lenti-X™ 293 T with either 2nd generation lentiviral packaging system (pMD2.G #12259 and psPAX2 #12260, Addgene) or the 3rd generation Virapower system (#K497500, Invitrogen), together with the plasmid of interest, using TransIT-LT1 reagent according to the manufacturer's protocol (#2304, Mirus). Supernatants were collected after 48–72 hrs and sterile-filtered (0.45µm filters, #SLHV033RS, Merck/Millipore). Target cells were infected with viral supernatants, supplemented with 3.5 µg/ml Polybrene Transfection reagent (#TR-1003-G, Merck/Millipore). After 48 hrs, transduced cells were selected by the addition of 3 µg/ml Puromycin (#P9620, Sigma), for another 48 hrs.

## Generation of Cas9 expressing lines

Subconfluent NF2-WT and NF2-KO#1 cells were infected in the presence of 6 µg/ml Polybrene with lentiviral particles harboring Lenti-Cas9-2A-Blast (#73310, Addgene). 48 hrs postinfection, cells were treated with 5 µg/ml Blasticidin for 72 hrs and single clones were obtained via cell dilution, in order to achieve homogeneous Cas9 expression levels in the selected lines for the Synthetic Lethality screen. Cas9-expressing lines were maintained in culture in the presence of 5 µg/ml Blasticidin (A11139-03, Gibco). Prior to the synthetic lethality screen, Blasticidin was removed from the cultures.

## Synthetic lethality screen

The genome-wide pooled synthetic lethality screen was performed using the Toronto CRISPR Human Knockout Library – TKO v3[18] which was obtained from Addgene. The screen protocol was based on[46] with minor adaptations: Subconfluent NF2-WT/Cas9 and NF2-KO/Cas9 cell lines were infected in the presence of 6 µg/ml polybrene (#TR-1003-G, Merck Millipore) with the 90k TKO sgRNAs library at a multiplicity of

infection (MOI) of 0.3, to achieve 500-fold coverage (individual sgRNA-editing events represented in 500 cells). The next day, Puromycin-containing medium (4 µg/ml) was added to the infected cells for 48 hrs. After selection, a portion of the puromycin-resistant population from each cell line was harvested for freezing and sequencing (T0 start point), and another portion was seeded for further culture expansion. Genomic DNA from cell pellets of both start-point (T0) and end-point (T=passage 7) were extracted using the QIAamp DNA Blood Maxi kit (#51192, Qiagen). To amplify the sgRNA sequences, a total of 140 PCR reactions were performed using 1 µg of genomic or plasmid library DNA, Q5 Hot Start HF polymerase (#M0493L, NEB), and primers harboring the Illumina TruSeq adapter sequences. PCR products were purified using DNA Clean and Concentrator TM-100 (#C1016-50, Zymo Research) and MagSi-NGSprep Plus beads (#SL-MDKT-01500, Steinbrenner). Sample concentrations were measured using the Qubit HS DNA Assay (#Q32851, Thermo Fisher). Library amplicon size was verified using the DNA High Sensitivity Assay on a BioAnalyzer 2100 (Agilent) and then sequenced on a NextSeq (Illumina) with 75 bp single-end sequencing and the addition of 25% PhiX control v3 (Illumina).

## Lentiviral plasmid cloning

For validation of the Synthetic Lethality screen results, sgRNAs targeting G6PD, ACSL3, and control AAVS1 (oligo sequences provided in Suppl. Table 2) were cloned into the lentiviral PLCKO vector (#73311 Addgene) according to the established protocol[47]. For the inducible knockdown system, shRNAs sequences targeting G6PD, ACSL3 and control RLuc (oligos provided in Suppl. Table 2) were cloned into the lentiviral Tet-pLKO-puro vector (#21915, Addgene) according to the standard protocol[48].

## Schwann cell cancer xenograft experiment

All animal experiments were done in accordance with the guidelines of the responsible national authority, and with the approval of the local Governmental Authority for Animal Experimentation (Regierungspräsidium Karlsruhe, Germany, license 35-9185.81/G-30/20). The maximal tumour size/burden (1.5 cm in diameter or 1.5 cm³ in volume) was not exceeded. Mice were maintained in a 12 hrs light-dark cycle with unrestricted Kliba diet 3307 and water. They were kept in individually ventilated cages at a temperature of about 22 °C and a humidity of 45–65%. Mice were bred inhouse. After adaptation, female mice were randomized according to age. In half of the mice, for knockdown induction, doxycycline (#D9891, Sigma; 1 mg/ml) was provided via drinking water supplemented with 5% saccharose three days prior to cell transplantation and was continued throughout the experiment. Controls received drinking water with 5% saccharose. Under isoflurane inhalation anesthesia (1–1.5% in air, 0.5 L/min), $10 \times 10^6$ cells suspended in 200 µl of a 1:1 (v/v) mix of DMEM (#41965-062, Gibco)/Matrigel (growth factor reduced, Corning #3821.00.00) were injected subcutaneously into the right flank of $8 \pm 1$ week-old female NOD SCID gamma (NSG) mice that were recruited from the Center for Preclinical Research, DKFZ, Heidelberg. Tumor volume was measured with a caliper up to three times a week and calculated according to the formula: V = (length (mm) x width (mm)²)/2. The weight of mice was documented once weekly. Necropsies were taken when one tumor diameter reached 1.3 cm or when any other pre-defined humane endpoint was reached. Animal well-being was monitored regularly, and animals were euthanized if any humane endpoints were reached in accordance with the approved license.

## Cell proliferation assays

Real-time cell growth of semi-confluent cells was monitored using the xCELLigence DP System (OLS, OMNI Life Science). 10.000 cells/well were seeded on E plate L8 PET (0.64 cm²) and impedance-based real-time proliferation was assessed over 96 hrs (96 sweeps with 1 hr

interval). In the xCELLigence system, impedance correlates with cell number if other cell properties stay equal. Images of fully confluent cultures (96 hrs) were taken using a 4x objective with a Leica Microscope (DMIL LED FLUO, Leica Microsystems). Proliferation curves of sub-confluent cells (2500 cells/0.34c m²) were measured with CellTiter-Glo (CTG) viability assay (#G9242, Promega), which quantifies total ATP in a well, which correlates with cell number.

## Crystal violet staining

To study the effect of G6PDi-1 on Schwann cell proliferation over several population doublings, 200.000 cells/well were seeded in a 6-well plate and passaged 1:4 every 48 hrs, up to 3 passages. To compare the proliferation of NF2-/+ cells vs NF2-WT and NF2-KO cells, 100.000 cells/well were seeded in a 12-well plate in biological triplicates. Cells were fixed for 20 min at room temperature with ice-cold Methanol. Afterwards 2xPBS washes were performed and fixed cells were stained with 0.5% crystal violet (#V5265, Sigma), diluted in 20% methanol/ddH₂0 for 20 min. After staining, extensive washes were performed with ddH₂0 and plates were allowed to dry. For quantification analysis, 10% acetic acid was added on top of the fixed cells and the Crystal Violet solution was measured at OD590 nm, using a spectrophotometer (Biospectrometer, Eppendorf).

## Validation of synthetic lethality

Synthetic lethality associations of NF2/G6PD and NF2/ACSL3 were evaluated via CellTiter-Glo (CTG) viability assay (#G9242, Promega) and CellTox cell death assay (#G8742, Promega). After Puromycin selection, infected cells were seeded in 96-well plates (2500 cells/well) and allowed to attach for 16 hrs. Cell number evaluation was performed using the automated cell counter TC20 (Biorad). Samples were analysed for cell viability and cytotoxicity at the selected endpoint ($T_{end}$= 96 hrs/4 days postseeding). Likewise, for experiments including treatment with inhibitors, 2500 cells were seeded in 96-well/plates (2500 cells/well). 16 hrs postseeding, cells were treated with the inhibitor and CTG and CellTox assays were performed 4 days post-treatment. Measurement of luminescence was done with TriStar Multimode Reader (LB 942, Berthold) and CellTox generated fluorescence was measured with Spectrostar Omega plate reader (BMG Labtech: Ex 485-12 nm/ Em 520 nm). Induction of shRNA expression was performed by treating cells with 1 µg/ml doxycycline hyclate (#D9891, Sigma) and the endpoint was at 96 hrs post doxycycline addition.

## Immunoblot analysis & antibodies

Protein extraction was performed with 1%SDS in PBS, supplemented with 1x protease and phosphatase inhibitor (#04693159001 and #4906837001, respectively, both from Roche). Cell lysates were sonicated for 10 sec to shear DNA (12% amplitude, Branson Digital Sonifier W-250D) and boiled for 3 min at 95 °C. Protein concentration was determined by BCA protein assay (Thermo Fisher Scientific) and colorimetric analysis was performed with a Spectrostar Omega plate reader (BMG Labtech, OD₅₆₂nm). 10–20 µg of total cell lysates were loaded on SDS-Page gels and transferred to Amersham nitrocellulose membranes (Merck). Description of all antibodies and dilutions used in this study are described in the Supplementary Table 1. Chemiluminescence was recorded with the Chemidoc Imager (Bio-Rad) and quantified using Image Lab software (Bio-Rad).

## RNA extraction, quantitative RT-PCR and RNAseq

For qRT-PCR, total RNA was extracted with TRIzol reagent (#15596026, Invitrogen) following manufacturer's instructions. cDNA was synthesized with the MaximaH minus Reverse Transcriptase (#EP0753, ThermoScientific) using 2 µg of total RNA as a template. Quantitative PCR was performed using Maxima SYBR Green/Rox (Fermentas), normalised to Rpl13a. The annealing temperature was set at 60 °C. Oligo primer sequences are provided in Suppl. Table 2. For RNAseq, total RNA was isolated using the RNeasy Mini Kit (#74106, Qiagen). Deep sequencing library preparation and data analysis was performed as previously described[49]. All sequencing data have been deposited at NCBI Geo (GSE219141). Gene enrichment analysis was done with ShinyGO software[17].

## RNA interference (siRNA)

Human Schwann cells were seeded at a density of $2 \times 10^5$ cells per well on six-well plates. The next day, cells were transfected with Lipofectamine RNAiMAX (#13778-500, Invitrogen) with a final concentration of 20 nM siRNA per well. For subsequent expression analysis, cells were lysed 96 hrs post-transfection. The following siRNAs were used (sequences provided on the supplementary table with oligos):

siRLuc: P-002070-01-50 (individual, ThermoFischer)
siNF2#1: siGenome D-003917-18-0002 (individual)
siNF2#2: siGenome D-003917-19-0002 (individual)
siG6PD: OnTarget plus LQ-008181-02-0002 (pool of 4)
siACSL3: OnTarget plus LQ-010061-00-0002 (pool of 4)

## Enzymatic activity assays and metabolite levels

G6PD activity was assayed using a commercially available kit, according to the manual's instructions (#MET-5081, Cell Biolabs) and samples were analysed with a Spectrostar Omega plate reader (BMG Labtech, OD₄₅₀nm). ME1 activity was performed on fresh cytosolic extracts as follows. The cells were collected and resuspended in 100 mM Tris-HCl lysis buffer containing 0.02% Digitonin to release cytosolic content but not mitochondrial content. The suspension was incubated on ice for 10 min and then centrifuged at 10000 g for 10 min at 4 °C. The supernatant containing the cell cytosolic extract was collected. The extracts were added to a reaction solution containing 100 mM Tris-HCl, 1 mM MnCl₂, 1 mM NH₄Cl, 100 mM KCl, and 1 mM NADP⁺ final concentrations. After adding 10 mM L-malate, the reaction plate was mixed by shaking briefly and immediately read as fluorescent intensity (Ex 355-20/Em LP460) for 1 hour with data collected every minute. The wells without L-malate were used as a blank. The data, subtracted from the blank data and normalized to protein concentration, were plotted and the slopes quantified to determine the relative cytosolic ME1 activity. Oxidized vs reduced NADP levels were assessed using NADP/NADPH Quantitation kit (#MAK038, Sigma) and glutathione levels were evaluated with GSH/GSSG-Glo™ Assay (#V6611, Promega).

## FACS

For the assessment of reactive oxygen species (ROS) levels, the general ROS indicator CM-H2DCFDA was used (#C6827, ThermoScientific) and for lipid ROS levels BODIPY™ 581/591 C11 (#D3861, ThermoScientific). Subconfluent cells were cultured overnight in standard conditions, and 30 min before FACS analysis, cells were incubated with 2.5 µM of the appropriate cell-permeant ROS indicator. Oligomycin treatment (5 µM for 30 min) was used as a positive control for ROS generation. FACS analysis was performed with Guava easyCyte HT (Millipore), using BlueV (Ex 450/45) and Yellow G (Em 575/25) lasers. Assessment of G6PDi-1-induced cell death was done with propidium iodide staining (#P4170, Sigma). FACS analysis was done using BlueV (Ex 450/45) and Red G (Em 695/50) lasers.

## Soft agar assay

Soft agar assay was performed using the cell transformation assay kit (#ab235699, Abcam). 10.000 cells were seeded on a soft agar matrix, following instructions of the kit, and colonies were visualized after 7 days with a Leica Microscope (DMIL LED FLUO, Leica Microsystems)

## Statistical analysis

Statistical analyses were done using GraphPad Prism 9 Software. The tests used to assess statistical significance are detailed in the figure

legends. We excluded one replicate of the experiment in Fig. 4j on request of Reviewer #2. Otherwise, no replicates were excluded.

## Figure preparation

Figures were prepared using Affinity Photo (https://affinity.serif.com/en-gb/).

## Data availability

The NGS sequencing data generated in this study have been deposited at NCBI Geo under accession code GSE219141. Source data are provided with this paper.

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

## Acknowledgements
We thank Dr. Margaret Wallace (University of Florida) and Dr. Valerio Magnaghi (University of Milano) for generously providing us with the immortalized human Schwann cell line (ipn02.3 2λ) and HEI-193 cells, respectively. We thank the DKFZ Genomics and Proteomics Core Facility for NGS-sequencing.

## Author contributions
A.K., R.V., Y.Z., G.A., S.L., and K. M.-D. performed experiments. A.K., R.V., Y.Z., G.A., S.L., K. M.-D., M.B. and A.A.T. designed experiments, analyzed data, and wrote the manuscript.

## Funding

## Competing interests
The authors declare no competing interests.
