## [Peer Review File · Nature Communications]

G6PD and ACSL3 are synthetic lethal partners of NF2 in Schwann cellsREVIEWER COMMENTS

Reviewer #1 (Remarks to the Author):

Kyrkou and colleagues report on the synthetic lethal interaction between loss of NF2 and G6PD or ACSL3. They discovered G6PD or ACSL3 in a genome wide CRISPR screen in vitro for synthetic lethals of NF2, and then used a battery of assays to validate and study the synthetic lethal interactions. Importantly they used an NF2 mutant Schwann cell line and its syngeneic wt counterpart for their assays. They find that the double KO triggered cell death through the activation of oxidative stress/ROS dependent cell death. The data are of high quality and the paper is very well presented.

I only have one question: the analysis of gene expression identified several pathways that are mis-regulated in NF2 mutant versus wt cells. NF2 is considered to be an upstream regulator of the Hippo pathway, although alternative models have been proposed. The authors detected Hippo signaling in their GO term analysis, but it was not the top hit. Are these other pathways affected indirectly or does NF2 act in multiple pathways? This question is not directly relevant to the paper, which focusses on NF2 and not on the Hippo pathway but given the broad activation of Yap and Taz in human cancer it would be interesting to know whether the synthetic lethal interactions may also be observed in other types of cancer. The authors may want to comment on that in any case.

Reviewer #2 (Remarks to the Author):

The manuscript "G6PD is a synthetic lethal partner of NF2 in Schwann cells" by Kyrkou et al. reports on the results of a genome-wide CRISPR/Cas9 screen identifying synthetic-lethal genes that cause death of Schwann cells with NF2 mutations but not Schwann cells with wildtype NF2. From the results of this screen, G6PD is identified as a potential therapeutic target because NF2 mutant Schwann cells also have low levels of ME1 and are thus susceptible to oxidative stress due to low NADPH. The authors conclude that G6PD could be a good pharmacological target for NF2.

The rationale for targeting G6PD and, more broadly, the study itself is unclear. Patients with syndromic neurofibromatosis type 2 most often encode germline inactivating mutations in the NF2 gene. Thus, "selectively" targeting cells with NF2 mutations (even if only just Schwann cells with NF2 mutations) could theoretically lead to catastrophic ablation of myriad non-tumor, otherwise-normal cells throughout the human body. Perhaps this pharmacologic strategy would be less catastrophic in syndromic neurofibromatosis type 2 patients encoding mosaic inactivating mutations in the NF2 gene? Or perhaps in patients with non-syndromic Schwann cell tumors associated with NF2 inactivation? Or perhaps in syndromic neurofibromatosis type 2 patients if there is robust assurance that 1 remaining wildtype copy of NF2 is sufficient to prevent therapeutic vulnerability (if this is the case, it should be tested in the authors' experimental models)? One way or another, this critical oversight in the study rationale must be clarified and (if possible) corrected.

Although ACSL3 was also identified as a significant hit from the authors' screen, the investigation of this particularly synthetic lethal dependency does not appear to be a primary focus of the study and should perhaps be removed to improve the narrative. It is notable that the ACSL3 investigations are robust and extensive (lipids, peroxidation, etc.) and could be retained if the authors wanted to shift the title/focus of the study away from a specific hit (G6PD) and instead focused more broadly on the results of their screen.

It is confusing why ME1 is suppressed in Schwann cells. Is this an artifact of the authors' model system? Or is this more broadly true and relevant to Schwann cells in patients with neurofibromatosis type 2? More broadly, these concerns underscore the dearth of cell lines and model systems employed in this study. Can the authors validate G6PD or ACSL3 as vulnerabilities in other NF2-deficient models, using either Schwann cells, meningioma cells, or ependymoma cells? HUVEC cells are used for one experiment, but this model does not seem particularly relevant (and perhaps only amplifies concerns with the study rationale articulated above). Can primary

schwannoma cells be incorporated? A small amount of re-analysis of public data is reported (Fig. 4g), but, mechanistically, it remains a mystery why ME1 would be suppressed in Schwann cells. Can the authors design and perform mechanistic experiments to shed light on this area of confusion that would seem to be central to their primary hypothesis? Could over-expression of active or catalytically inactive ME1 to rescue (or not) the phenotypes observed be performed?

Beyond these more fundamental concerns, there are a series of perhaps more minor concerns that nevertheless dampen enthusiasm for the anticipated impact of this work:

1. The abstract over-simplifies the phenotypic consequences of NF2 inactivation, which includes formation of tumors arising from the meningeal lining or ependyma of the CNS. Moreover, it is untrue that "no pharmacological therapy is currently available for patients with NF2, as Avastin/bevacizumab has shown recent promise and approval for treating symptomatic vestibular schwannoma growth in patients with NF2.
2. The authors cite the prevalence of G6PD deficiency worldwide in the concluding remarks of their Introduction. What is the prevalence of G6PD deficiency in patients with syndromic NF2? If this is not known or relevant to the study itself, these speculative comments are recommended to be moved to the Discussion.
3. Can the authors clarify what other genes have been interrogated and/or identified as altered in ipn0.2 2l cells? This information is important for understanding how specific the reported phenotypes are for syndromic NF2, as opposed to schwannomatosis or sporadic Schwann cell tumors.
4. The blot in Fig. 1a should show a full range of molecular weights to confirm that NF2 has indeed been deleted from ipn0.2 2l cells, using antibodies that recognize either the N- or C-terminal end of the protein to assure complete ablation. Otherwise, a hypofunctional partial allele may remain. Alternatively, CRISPRi could be used to assure global transcript suppression. Indeed, the noise in the green dots in Fig. 1f suggests that there may be small residual NF2 activity in ipn0.2 2l cells.
5. How specific are the transcriptomic changes in ipn0.2 2l cells with NF2 suppression to the Schwann cell lineage? Choudhury et al. Nat Genet 2022 has reported that meningioma cells with/without NF2 suppression have very different gene expression programs from Schwann/schwannoma cells with/without NF2 suppression. Is this also true for ipn0.2 2l cells? Differential comparison between the data from Choudhury et al. and those reported here are encouraged.
6. Why were 7 cell passages used for the genome-wide CRISPR/Cas9 screen?
7. The data in Fig. 2a appear to be highly influenced (biased?) by 1 very high value in the NF2-KO#1 control condition. This experiment must be repeated with more replicates to ensure statistical significance (to say nothing of biological significance) are retained. The same is true of Fig. 4d and Fig. 4j
8. The immunoblot in Fig. 2c does not support the quantification of cell death in Fig. 2b. Namely, there is more cleaved caspase 3 in the NF2-KO#1 control condition of Fig. 2c, but cleaved caspase 3 is increased in the NF2-KO#1 experimental condition of Fig. 2b. This problem appears to also be present in Fig. 2i and Fig. 2j.
9. Extended Data Fig. 2 is problematic. The decrease in cell count in ED Fig. 2a-b is perhaps 25%, perhaps half of what is shown in the main figure. There are no error bars or statistics in ED Fig. 2d. Claims of growth factor stimulation with 5% vs 10% serum are suspect and should be repeated in serum-free conditions.
10. The authors report generating xenograft tumors with NF2-KO cells, but do not show images or histology from these reported xenografts. Both are absolutely essential, along with graphs showing tumor initiating capacity and growth rate over time.

Reviewer #3 (Remarks to the Author):

This study by Kyrkou et al, aims at identifying candidate genes that are synthetic lethal to NF2 mutations to specifically eliminate NF2 mutant cancer cells and spare normal cells. NF2 encodes for the protein Merlin, a tumor suppressor known to regulate cell-cell adhesion and activate the Hippo pathway through several mechanisms. Merlin is known to be particularly expressed in Schwann cells and to activate the Hypo pathway, MST1/2 and LST1/2 kinases, and promote the phosphorylation of the cotranscription factors YAP/TAZ and their

inactivation/sequestration in the cytoplasm. Neurofibromatosis Type II (NFII) is a rare genetic disorder that can be inherited or can appear as a de novo mutation. In human, mutations in the NF2 gene lead to the production of abnormal shortened version of the Merlin protein and cause Schwannoma. Tumors grow along the vestibulocochlear nerves, a peripheral cranial nerve that controls balance and inner ear functions. Those tumors are usually benign, but cross-sectional studies reveal that if NFII is associated with intracranial meningioma or if diagnosed at young age, NFII becomes life-threatening, and the risk of mortality increases presumably because tumor growth is generally more rapid in young patients. Because there are no approved chemotherapies for NFII, patients endure surgical removal of the tumors with high risk to lead to nerve damages, post operative complications and death.

Finding therapeutic approach for this very serious life-threatening disease is very significant.

The authors find one potential target that can specifically reduce cell viability of mutant SC lacking NF2. However, the approach, conclusions, discussion, and technical rigor of this study reduce considerably the enthusiasm to accept this study for publication in nature Communication.

My main problem with this approach is that single targeting synthetic lethal gene without analyzing how Schwann cells compensate for this loss is too risky. The effect obtained might not be the direct consequences of the loss of G6PD but consequential to the dysregulation of other unknown genes when G6PD is lost.

In addition, while G6PD reduces mutant SC proliferation or increase their cell death, there is a large proportion of mutant SCs still present. Then, the first possible effect of reducing the oxidative stress regulators such as G6PD in the Schwannoma cells is to create more genetic alterations and potentially more aggressive cancerous cells. This point is not discussed and is a big flaw in this study, unless the authors run an RNA-seq in their NF2-KO SC treated with G6PD shRNA and prove that the treatment is not worse than no treatment.

Another problem in this study is that while the authors claim that G6PD and ME1 redundantly generate cytosolic NADPH needed by cells to fight oxidative stress and that lack of either one is compatible with cell viability, it is assumed that all Schwann cells will express the same level of ME1. While it is well known that there is a large heterogeneity in SC gene expression and protein production, molecular pathway... there no study showing that all Schwann cells in the body express sufficient level of ME1 to compensate for the lack of G6PD. Thus it is unclear if all non-cancerous Schwann cells won't be affected by a reduction of G6PD expression and generate SC death in other part of the peripheral nervous system.

While the approach of finding a synthetic lethal gene using CRISPR/Cas9 in NF2 mutant SCs is novel, the execution is poor and conclusions made not well supported by the data presented.

While the number of single genes in the human CRISPR/Cas9 library being tested is unknown (the authors omitted to give us a number), the fact that a single hit is found among x genes tested might simply mean that the synthetic lethal gene approach is not appropriate for NFII.

Also the gene candidate the authors are proposing regulate the oxidative stress cascade, pathway that has already been recently published by the Parkinson's lab in the journal Brain in 2022, that already described pharmacological approach to inhibit this pathway.

The methods used are incomplete and do not enable to reproduce the results.

The authors omitted to describe from which nerves the cells they used in their screening were derived from. It appears that ipn02.3 2λ are derived from human sural nerves. Based on the known heterogeneity of Schwann cell subtypes within the peripheral nervous system, could the authors give a rationale on why using Schwann cells derived from sural nerves (ipn02.3 2λ) is a good model to screen for synthetic lethal gene in NFII, a cranial nerve disease? Are sural nerves affected in NFII patients? If not, the authors need to discuss how molecular regulations of the SCs from sural nerves are comparable to SCs from the vestibulocochlear nerves (ex: regulation of their oxidative stress level, ME1, AKR1C3, GPX4 expression levels...).

Overall, there is a lack of explanation in the technique used or results obtained that does not help the reader follow the logic of the result interpretation and conclusions.

Here are some examples of lack of clarity:

-It is unclear why both Fig.1b and 1c are presented, are they both showing the same results, an increase in cell numbers when NF2 is lost? What is Cell Titer Glo compared to Cell Index

iCelligence? Are they both indicating cell survival or cell proliferation? material and methods neither legend and results have a clear description about this.

-Why the relative cell number is lower in NF2-KO/Cas9 versus NF2-KO#1 and NF2-KO#2? Performing a CRISPR/cas9 for NF2 in NF2-KO mutant cells should not impact cell number unless the CRISPR/cas9 for NF2 has also off targets. The authors need to provide an explanation for this discrepancy.

- 'This identified many genes required for cell viability in both cell lines ("essential genes", Fig. 1f).' It is unclear if the following sentence means that all the other genes presented in Fig1f are essential genes except for G6PD and ACSL3.

- For the genome-wide CRISPR/Cas9 library how many gRNAs targeting how many protein coding genes, and which control were used to control for the rate of infections? It is important to discuss the number of positive hits (??) versus 'x' number of targets tested, this could mean that NF2 mutant Schwann cells have low level of sensitivity to synthetic lethal gene approach.

- 'We excluded from further consideration any gene that causes increased proliferation of NF2-WT cells, since these are potentially tumor suppressors and could lead to tumors if targeted pharmacologically in patients. Likewise, we excluded any gene that causes reduced cell numbers in NF2-WT cells since targeting them pharmacologically could be toxic.'

Where is the list of those genes?

- 'NF2-WT and NF2-KO cells at low density so that both lines showed a similar ~5-fold increase in cell number over the course of 4 days when transduced with negative-control sgRNA (Fig. 2a).'

Please clarify what was used as negative control sgRNA and how the infection efficiency was controlled.

- 'This was due at least in part to a significant, 6-fold increase in cell death upon G6PD loss-of-function in NF2-KO cells, assessed by CellTox (Fig. 2b).'

It is unclear if the Cell Tox was also done after 4 days in culture so that the corresponding reduce number of cells in NF2KO G6PD Fig2a can really be correlated with the 6 fold cell death fig 2b.

Concerning the xenograft experiment

-Because there is no explanation why RLuc is used as a negative control and there is no quantification of luciferase expression it is unclear what the control RLuc is used for in Figure2? please clarify.

-There are no representative images of the cells post subcutaneous xenograft that have been used for all the quantifications presented in Figure2. It is unclear how those quantifications were done in Figure 1 and 2, please also provide a detailed explanation of the methods used for quantifications (ex: just citing iCelligence or Cell Titer Glo is not sufficient).

-It is unclear why there is investigation on why the loss of ACSL3 lead to NF2-KO SC death while why the mechanism of G6PD loss of expression that lead SC death is not being explored while at the end of the study the authors clearly mention that the more targetable gene to treat NFII is G6PD. "Overall, to us G6PD appears most promising because a G6PD inhibitor is likely to have a mild side-effect profile if dosed appropriately".

It is not clear why a subcutaneous xenograft of the NF2-KO cells was used instead of onto the cochleovestibular nerve, knowing that the Hippo pathway and mechanotransduction respond differently depending on the cellular environment (abdomen cavity vs cochleovestibular region).

The manuscript has big flaws and misses the opportunity to cite key publications in the field of Schwannoma cells Hippo pathway and YAP TAZ. Consequently, inaccurate conclusions are made. For example, saying that YAP and TAZ are required for myelin maintenance is wrong since this is still debated in the field (Grove versus Jeannette et al, 2020).

The authors never use Merlin in their manuscript, which is the protein coded by NF2.

Overall, this shows a lack of rigor in preparing the manuscript and lack of knowledge of the field. Schultz et al., 2013, Mindos et al., 2017, Jeannette et al, 2020, Feltri et al., 2021, Labara et al., 2022 ...

Reviewer #1

Kyrkou and colleagues report on the synthetic lethal interaction between loss of NF2 and G6PD or ACSL3. They discovered G6PD or ACSL3 in a genome wide CRISPR screen in vitro for synthetic lethals of NF2, and then used a battery of assays to validate and study the synthetic lethal interactions. Importantly they used an NF2 mutant Schwann cell line and its syngeneic wt counterpart for their assays. They find that the double KO triggered cell death through the activation of oxidative stress/ROS dependent cell death. The data are of high quality and the paper is very well presented.

We thank the reviewer for the kind and supportive words.

I only have one question: the analysis of gene expression identified several pathways that are mis-regulated in NF2 mutant versus wt cells. NF2 is considered to be an upstream regulator of the Hippo pathway, although alternative models have been proposed. The authors detected Hippo signaling in their GO term analysis, but it was not the top hit. Are these other pathways affected indirectly or does NF2 act in multiple pathways? This question is not directly relevant to the paper, which focusses on NF2 and not on the Hippo pathway but given the broad activation of Yap and Taz in human cancer it would be interesting to know whether the synthetic lethal interactions may also be observed in other types of cancer. The authors may want to comment on that in any case.

The reviewer raises 2 interesting questions:

1. is the ME1 down-regulation due to hippo/YAP/TAZ signaling? and
2. is the NF2/G6PD synthetic lethality observed in other cells lines when YAP/TAZ are activated?

To address question #1 we tried several things:

- As the reviewer points out, besides the Hippo/YAP/TAZ pathway, NF2 also regulates the PI3K/Akt/mTOR and Src/MAPK pathways. Indeed, we do see elevated pAkt but not elevated pERK, in our NF2-KO cells (Fig. 1A). Therefore, to test whether the drop in ME1 levels in NF2-KO cells is due to activation of the PI3K pathway, we inhibited PI3K pharmacologically with LY294002 for 24h (Reviewer Figure 1). However, this did not restore ME1 levels in NF2-KO cells to those of NF2-WT cells, indicating that the drop in ME1 is not due to PI3K.

Reviewer Figure 1: Inhibition of the PI3K pathway with LY294002 does not restore ME1 levels in the NF2-KO cells back to wildtype levels.

This suggests that regulation of ME1 by NF2 probably goes via the Hippo/YAP/TAZ pathway.

- The YAP/TAZ co-activators can interact with several different transcription factors to regulate gene expression: the TEADs (which are the canonical interacting partners), AP-1 transcription factors, CDK7, p73, RUNX1/2, etc. (PMID 34782888). To abolish all of these possible YAP/TAZ transcriptional complexes, we tried to make YAP/TAZ double knockout (DKO) Schwann cells. Although we were able to obtain the single KOs, we did not get any DKO, strongly suggesting that the DKO is lethal.

Instead, we have started targeting the individual YAP/TAZ-interacting proteins. We pharmacologically inhibited the TEADs with VT-104 or K-975 and then assayed expression of ME1 and the YAP/TAZ target gene ANKR1 by Q-RT-PCR (Reviewer Fig. 2). As expected, we see that in the NF2-KO cells ME1 mRNA levels are lower than in NF2-WT cells, while ANKR1 mRNA levels are 1000x higher than in NF2-WT cells. Inhibition of the TEADs, surprisingly, does not bring ANKR1 levels back down to the level of NF2-WT cells. In NF2-KO cells + VT-104, ANKR1 levels are still 500-fold higher than in NF2-WT cells. One option is that the TEAD inhibitor is not very effective. However, we see this phenotype with high concentrations of VT-104, and also with K-975. Another option, which we favor, is that YAP/TAZ regulate gene expression in Schwann cells in large part by binding additional transcription factors. Paralleling the results obtained with ANKR1, the TEAD inhibitors do not restore ME1 levels in the NF2-KO cells back to NF2-WT levels.

Reviewer Figure 2: Inhibition of TEADs with VT-104 does not rescue the altered gene expression seen in NF2-KO cells back to NF2-WT levels.

In sum, we think the most likely explanation is that ME1 is regulated via YAP/TAZ but not via the TEADs. This is a topic we are currently working on, but will probably take an entire project to clarify in a solid way.

To address the 2nd question, we generated by CRISPR/Cas9 four additional NF2 knockout lines of different origins, in addition to the HUVECs which we had in the original submission, and tested all 5 for synthetic lethality with G6PD (new Suppl. Fig. 7A-E). In none of these 5 lines that we tested, however, does G6PD show synthetic lethality with NF2. This likely reflects tissue-specific biology. In HeLa and endothelial HUVEC cells, loss of NF2 does not lead to YAP activation (lower phosphorylation), which is likely the explanation for the lack of synthetic lethality. Instead, in U2OS osteosarcoma cells, HCT116 colorectal carcinoma cells, and human fibroblasts, loss of NF2 does lead to YAP activation but not loss of ME1. In sum, it looks like the synthetic lethality between NF2 and G6PD is specific to glia or Schwann cells.

We have included these new data in the Results section of the manuscript.

Reviewer #2

The manuscript "G6PD is a synthetic lethal partner of NF2 in Schwann cells" by Kyrkou et al. reports on the results of a genome-wide CRISPR/Cas9 screen identifying synthetic-lethal genes that cause death of Schwann cells with NF2 mutations but not Schwann cells with wildtype NF2. From the results of this screen, G6PD is identified as a potential therapeutic target because NF2 mutant Schwann cells also have low levels of ME1 and are thus susceptible to oxidative stress due to low NADPH. The authors conclude that G6PD could be a good pharmacological target for NF2.

The rationale for targeting G6PD and, more broadly, the study itself is unclear. Patients with syndromic neurofibromatosis type 2 most often encode germline inactivating mutations in the NF2 gene. Thus, "selectively" targeting cells with NF2 mutations (even if only just Schwann cells with NF2 mutations) could theoretically lead to catastrophic ablation of myriad non-tumor, otherwise-normal cells throughout the human body. Perhaps this pharmacologic strategy would be less catastrophic in syndromic neurofibromatosis type 2 patients encoding mosaic inactivating mutations in the NF2 gene? Or perhaps in patients with non-syndromic Schwann cell tumors associated with NF2 inactivation?

Or perhaps in syndromic neurofibromatosis type 2 patients if there is robust assurance that 1 remaining wildtype copy of NF2 is sufficient to prevent therapeutic vulnerability (if this is the case, it should be tested in the authors' experimental models)? One way or another, this critical oversight in the study rationale must be clarified and (if possible) corrected.

The reviewer raises a very important point which we failed to address in the original submission. Since previous studies had shown that very little NF2 protein is sufficient to restore full NF2 function in cells (PMID 24336571, 21383154), we had assumed that heterozygous NF2-/+ cells would behave like wildtype cells, but we had not tested this! We now provide data explicitly testing this, and find that as the reviewer suggests, 1 remaining wildtype copy of NF2 is sufficient to prevent therapeutic vulnerability:

We generated by CRISPR/Cas9 a Schwann cell line that has one allele with an out-of-frame mutation, and the other allele with an in-frame loss of 3 nt, which leads to a protein lacking 1 amino acid (Suppl. Fig. 2H-I). These cells, therefore, are roughly heterozygous. (If anything, they have a bit less NF2 activity than an NF2-/+ heterozygous line because the remaining protein is missing 1 amino acid which could lead to reduced NF2 activity.) Indeed, in Suppl. Fig. 2I one can see that these cells have low NF2 protein levels. In agreement with previous publications (PMID 24336571, 21383154), however, these cells proliferate like NF2-WT cells (Suppl. Fig. 2J). Likewise, these cells do not show synthetic lethality with G6PD (Fig. 2L).

We also provide in new Suppl. Fig. 7A-E data where we generated several additional NF2 knockout lines of different origins, in addition to the HUVECs which we had in the original submission, and tested synthetic lethality with G6PD (new Suppl. Fig. 7A-E). In none of these 5 lines that we tested, however, does G6PD show synthetic lethality with NF2. This likely reflects tissue specificity in the biology. For instance, loss of NF2 does not lead to loss of ME1 in HCT116 cells or fibroblasts. In sum, it looks like the synthetic lethality between NF2 and G6PD is specific to glia or Schwann cells.

In sum, we think catastrophic ablation of non-tumor cells can most likely be ruled out.

Although ACSL3 was also identified as a significant hit from the authors' screen, the investigation of this particularly synthetic lethal dependency does not appear to be a primary focus of the study and should perhaps be removed to improve the narrative. It is notable that the ACSL3 investigations are robust and extensive (lipids, peroxidation, etc.) and could be retained if the authors wanted to shift the title/focus of the study away from a specific hit (G6PD) and instead focused more broadly on the results of their screen.

We thank the reviewer for this comment. We think it would be a pity to remove these data because we also believe, as the reviewer writes, that the ACSL3 data are robust, and the synthetic lethality effects are even stronger than with G6PD. In fact, ACSL3 and G6PD are both related to oxidative stress, suggesting common underlying biology. (ACSL3 counteracts lipid peroxidation and ferroptosis.) We have therefore changed the title to also include ACSL3, and we have significantly re-ordered and re-written the manuscript to highlight the commonalities between G6PD and ACSL3 in terms of oxidative stress. We think this has significantly improved the manuscript and in particular the presentation of the underlying logic.

It is confusing why ME1 is suppressed in Schwann cells. Is this an artifact of the authors' model system? Or is this more broadly true and relevant to Schwann cells in patients with neurofibromatosis type 2? More broadly, these concerns underscore the dearth of cell lines and model systems employed in this study. Can the authors validate G6PD or ACSL3 as vulnerabilities in other NF2-deficient models, using either Schwann cells, meningioma cells, or ependymoma cells? HUVEC cells are used for one experiment, but this model does not seem particularly relevant (and perhaps only amplifies concerns with the study rationale articulated above). Can primary schwannoma cells be incorporated? A small amount of re-analysis of public data is reported (Fig. 4g), but, mechanistically, it remains a mystery why ME1 would be suppressed in Schwann cells. Can the authors design and perform mechanistic experiments to shed light on this area of confusion that would seem to be central to their primary hypothesis? Could over-expression of active or catalytically inactive ME1 to rescue (or not) the phenotypes observed be performed?

The reviewer raises multiple points, which we address sequentially:

"Is it an artifact?"

We do not think this is an artifact of the model system because we see it in multiple different setups where we knockout NF2 in Schwann cells:

- We see it in two independently generated ipn02.3 Δ NF2-KO cell lines, which were generated with two independent sgRNAs, targeting two different exons (KO#1 and KO#2 in Fig. 5D and Suppl. Fig. 1A). This excludes that the drop in ME1 is a clonal effect or due to an off-target effect of an sgRNA.
- We see it in a primary human Schwann cell line with a completely independent genetic background (Suppl. Fig. 4A). Please note that primary cells do not double enough times before becoming quiescent, so we could not generate a complete NF2-KO line via CRISPR/Cas9. As a result, we needed to use siRNAs, which do not deplete NF2 as efficiently as the sgRNAs. As mentioned above, even small amounts of NF2 are sufficient to provide the cell with enough NF2 activity. In this case, we achieve a

60%-70% knockdown efficiency, which is good enough to yield a drop in both ME1 and AKR1C3, but it's not as dramatic as in the ipn02.3 2λ cells with complete NF2 loss-of-function, which replicates better the patient condition.

- We see it in the vast majority of the primary patient schwannomas that we analyzed (Fig. 5H): 46 of the 58 schwannomas have lower ME1 than the mean of the control nerve samples, and all 13 out of 13 NF2 schwannomas have lower ME1 than the control mean. Hence, as the Reviewer asks, we do see it in Schwann cells of patients with neurofibromatosis type 2. This result is striking given that each of the samples has a different genetic background which therefore introduces significant variability, yet the drop in ME1 is visible. Also worth pointing out is that these tumor samples contain both Schwann cells forming the tumor and other non-Schwann cells, which reduces the magnitude of the difference in gene expression that can possibly be observed.

- We now also include in Suppl. Fig. 4 data from HEI-193 cells. This is a Schwann cell line from an NF2 patient with spontaneous bilateral vestibular schwannomas. The cells have a point mutation that causes a splicing defect in the NF2 transcript, and thereby a partial NF2 loss-of-function (PMID 17868749). As expected, these cells express NF2 'isoform 3' which runs lower than full-length NF2, and they have reduced YAP phosphorylation (Suppl. Fig. 4B). In addition, they have low levels of ME1 and AKR1C3 (Suppl. Fig. 4B).

- We now also tested another batch of primary human Schwann cells from a separate source compared to the one in Suppl. Fig. 4A. The same caveat holds true for these primary cells - since they double a limited number of times, we could not generate a complete NF2-KO line. Instead, we performed sequential infection with lentiviruses expressing Cas9, followed by a second infection with guide RNA targeting NF2, and lysed the pooled population 2 days after puromycin selection. Although the NF2 depletion is not great, there is nonetheless a drop in ME1 levels to 64% also in this case (Reviewer Figure 3).

In sum, we see a drop in ME1 levels upon loss of NF2 in Schwann cells in 18 different contexts. So we believe it is not an artifact.

Also worth mentioning is that ME1 is only one of several enzymes involved in fighting oxidative stress which drop upon loss of NF2 in Schwann cells: also AKR1C1, AKR1C2, and AKR1C3 drop (Fig. 5D, 5I, Suppl. Fig. 4A, Suppl. Fig. 4B). We have also carefully measured G6PD protein levels and activity, and they also drop upon loss of NF2 (Fig. 5D, 5F). So ME1 is not an 'outlier' but part of a more general pattern. It appears that loss of NF2 in Schwann cells causes changes in gene expression that render them sensitive to oxidative stress.

"There is a dearth of cell lines"

The reviewer points to an issue which unfortunately is not specific to our manuscript, but reflects the state of the field at the moment. There simply are not many human NF2-WT Schwann cell lines that are available and amenable:

- The ipn02.3 2λ Schwann cells from the Wallace lab which we use here are the 'standard' line that, to our knowledge, everyone in the field is using.
- We purchased primary human Schwann cells from Creative Bioarray. We were able to get these cells to grow and expand for even more passages than promised by the company (7 passages) and the data from these cells are in Suppl. Fig. 4A. (Note that nonetheless, although we tried, 7 passages are not sufficient to generate KO lines by CRISPR/Cas. Knockdowns with siRNAs are sufficient to do immunoblots, but the knockdown effect does not last long enough to do proliferation assays.)
- We purchased primary human Schwann cells from Sciencell, which grew sufficiently to enable us to look at a polyclonal population of cells that had been transduced with cas9 and an sgRNA targeting NF2 (but not to generate a clean, complete KO), and the data are in Reviewer Figure 3 above.
- Margaret Wallace also kindly provided us with the immortalized ipn97.4 line which she generated, but these cells do not expand well, so we were not able to use them for any experiments.
- We got immortalized Schwann cells from Ahmet Höke (PMID 21585251) but these cells did not expand in our hands (upon thawing the aliquot we received, the cells simply sat in the plate for days - they were senescent and had strange morphology).
- We purchased a newly generated immortalized NF2-WT Schwann cell line from Creative Bioarrays but they also do not grow well - we split them once after 10 days because they were growing so slowly, and then they stopped growing altogether.
- In an attempt to try additional NF2-WT Schwann cell systems, we purchased rat S42 Schwann cells from ATCC and generated NF2-KO cells by CRISPR/Cas9. Strangely, knockout of NF2 in these cells does not lead to YAP activation (ie reduced YAP phosphorylation) (Reviewer Fig. 4). This indicates that these rat cells do not recapitulate the regulation observed in human Schwann cells from patients, and do not constitute a good system for studying NF2 biology.

Reviewer Figure 4: YAP activity does not increase in rat S42 Schwann cells upon knockout of NF2, indicating that they do not constitute a good system for studying NF2 biology.

- The lab of Pi-Nan Liu published in 2022 new Schwann cell lines in a paper entitled "Novel patient-derived xenograft and cell line models for therapeutic screening in NF2-associated schwannoma". I emailed the senior author Pi-Nan Liu twice, and then the first author Fu Zhao twice, asking for these lines, and I never got a reply from either author.

Thus, as far as we know, we have now tested everything that is available in the field.

In sum, to address this concern, we have added data from two additional cell lines (HEI-193 and primary Schwann cells from Sciencell), and our results are supported with data from 58 patients. We have also added a section to the manuscript at the end of the Discussion "Limitation of our study" which mentions explicitly that the number of cell lines that we could test is limited.

"Does the synthetic lethality also occur in other cell types besides Schwann cells?"

To address this issue, we generated 4 additional NF2 KO lines in different cell types in addition to the HUVEC cells (HeLa, U2OS, HCT116, and human fibroblasts) and, as described above, we did not see synthetic lethality, suggesting that the synthetic lethality is specific to Schwann cells.

As suggested by the reviewer, we also tested meningiomas, and these data look potentially promising. Firstly, we analyzed publicly available patient data. Tsitsikov et al. (PMID 36937440) performed gene expression analysis by RNA-seq on 20 meningiomas from patients - 10 with NF2 gene inactivation and 10 not. This shows a statistically significant drop in ME1 in the NF2 mutant meningiomas compared to the meningiomas without NF2 mutation (Reviewer Fig. 5).

Reviewer Figure 5: ME1 mRNA levels are reduced in NF2 mutant meningiomas. Re-analysis with GEO2R of data from PMID 36937440. **** $p \leq 0.0001$ by t-test.

We also found a second dataset from Clark et al (PMID 23348505) where meningiomas were categorized for NF2 status and gene expression analysis was performed by microarray. Note that microarray analysis usually compresses the dynamic range in the data. Also in this dataset there is a statistically significant drop in ME1 levels in the NF2 mutant meningiomas, although the apparent magnitude of the effect is small, likely due to the microarray.

Reviewer Figure 6: ME1 mRNA levels are reduced in NF2 mutant meningiomas. Re-analysis with GEO2R of data from PMID 23348505. *** $p = 0.0006$ by t-test.

We then obtained a meningioma cell line, IOMM-Lee, and tested synthetic lethality with G6PD. We first tested how IOMM-Lee cells compare to ipn02.3 Δ Schwann cells, and found that IOMM-Lee cells have significantly higher NF2 phosphorylation (which inhibits NF2 activity - PMID 15378014) and correspondingly low YAP phosphorylation (Reviewer Figure 7A). This indicates that IOMM-Lee cells already have very low NF2 activity and high YAP activity. Indeed, we generated NF2-KO IOMM-Lee cells by CRISPR/Cas9 and found that the NF2 knockout hardly has an effect on YAP phosphorylation (Reviewer Figure 7B), indicating that the parental IOMM-Lee cells already have an almost-complete NF2 loss-of-function.

Reviewer Figure 7: IOMM-Lee cells have low NF2 activity and die upon G6PD inhibition.

(A) Immunoblot comparing protein and phospho-protein levels in IOMM-Lee cells to Schwann cells. IOMM-Lee cells have high inhibitory NF2 phosphorylation and low inhibitory YAP phosphorylation.

(B) Knockout of NF2 in IOMM-Lee cells does not significantly decrease YAP phosphorylation, indicating that parental IOMM-Lee cells have very little NF2 activity.

(C) Both parental and NF2-KO IOMM-Lee cells die upon G6PD inhibition using 25 μM G6PDi-1.

We then looked at expression of other redox genes and found that IOMM-Lee cells have low IDH1, IDH2 and AKR1C3, but not low ME1 (Reviewer Fig. 7A). Either meningeal cells having higher ME1 expression than Schwann cells (note that the patient expression data in Reviewer Figs. 5 and 6 compare NF2 mutant meningiomas to other meningiomas, and not to Schwann cells) or this one cell line goes against the general trend. Nonetheless, IOMM-Lee cells are extremely sensitive to G6PD inhibition (Reviewer Figure 7C). We see no big difference between the NF2-KO and the parental IOMM-Lee (both die readily) in agreement with the fact that the parental line is already NF2 loss-of-function (Reviewer Fig. 7B). In fact, IOMM-Lee cells even die with 10 μM G6PD-i (not shown). This is different from all the other cells that we tested in Suppl. Fig. 7, which were not sensitive to 25 μM G6PD-i, indicating that meningioma may indeed be another indication where G6PD inhibition could be therapeutically useful.

(To extend these findings further, we also test immortalized meningeal cells from the Innoprot but they did not grow, and we tested primary meningeal cells from Sciencell, but we were not able to achieve efficient knockdown of NF2 using siRNA transfections or viral transductions). In sum, we believe an entire study should be done to properly assess whether G6PD inhibition would be a therapeutic approach to treat meningioma, which is why we are providing these intriguing initial data to the reviewer, but are not including them in the manuscript.

"What is the mechanism leading to low ME1 mRNA in Schwann cells upon loss of NF2?"

Reduced ME1 mRNA levels can be due to reduced transcription, reduced mRNA stability due to elements in the 3'UTR (e.g. microRNA target sites) or both. To test whether the 3'UTR of ME1 is contributing to this regulation, we cloned the ME1 3'UTR downstream of Renilla Luciferase (RLuc), and co-transfected it with a Firefly Luciferase (FLuc) control into NF2-WT and NF2-KO cells (Reviewer Figure 8). This is the standard test to assay whether there are destabilizing elements in a 3'UTR, such as miRNA sites.

Indeed, the ME1 3'UTR reduces RLuc levels in both NF2-KO cell lines, indicating that it contains some elements that are destabilized in KO cells compared to WT cells. Thus the 3'UTR contributes to the reduced ME1 mRNA levels in NF2-KO cells. This effect, however, is not as strong as the reduction in ME1 mRNA levels in NF2-KO cells, where they drop to roughly 10%. Hence, this indicates that reduced transcription likely also contributes.

To identify the transcriptional link from NF2 to ME1, we undertook two approaches: one that is 'top-down' (from NF2 down the signaling pathway) and one 'bottom-up' (from ME1 to regulatory transcription factors):

Top-down:

- NF2 can regulate several pathways: the Hippo/YAP/TAZ pathway, the PI3K/Akt/mTOR pathway, and the Src/MAPK pathway. We see elevated pAkt but not elevated pERK, in our NF2-KO cells (Fig. 1A). Therefore, this excludes effects via Src/MAPK. To test whether the drop in ME1 levels in NF2-KO cells is due to activation of the PI3K pathway, we inhibited PI3K pharmacologically with LY294002 for 24h (Reviewer Figure 9). However, this did not restore ME1 levels in NF2-KO cells to those of NF2-WT cells, indicating that the drop in ME1 is not due to PI3K.

Reviewer Figure 9: Inhibition of the PI3K pathway with LY294002 does not restore ME1 levels in the NF2-KO cells back to wildtype levels.

This suggests that regulation of ME1 by NF2 likely goes via the Hippo/YAP/TAZ pathway.

- The YAP/TAZ co-activators can interact with several different transcription factors to regulate gene expression: the TEADs (which are the canonical interacting partners), AP-1 transcription factors, CDK7, p73, RUNX1/2, etc. (PMID 34782888). To abolish all of these possible YAP/TAZ transcriptional complexes, we tried to make YAP/TAZ double knockout (DKO) Schwann cells. Although we were able to obtain the single KOs, we did not get any DKO, strongly suggesting that the DKO is lethal.

Instead, we targeted the individual YAP/TAZ-interacting proteins. We pharmacologically inhibited the TEADs with VT-104 or K-975 and then assayed expression of ME1 and the YAP/TAZ target gene ANKR1 by Q-RT-PCR (Reviewer Fig. 10). As expected, we see that in NF2-KO cells ME1 mRNA levels are lower than in NF2-WT cells, while ANKR1 mRNA levels are 1000x higher than in NF2-WT cells. Inhibition of the TEADs, surprisingly, does not bring ANKR1 levels back down to the level of NF2-WT cells. In NF2-KO cells + VT-104, ANKR1 levels are still 500-fold higher than in NF2-WT cells. One option is that the TEAD inhibitor is not very effective. However, we see this phenotype with high concentrations of VT-104, and also with K-975. Another option, which we favor, is that YAP/TAZ regulate gene expression in Schwann cells in large part by binding other transcription factors. Paralleling the results obtained with ANKR1, the TEAD inhibitors do not restore ME1 levels in the NF2-KO cells back to NF2-WT levels.

Reviewer Figure 10: Inhibition of TEADs with VT-104 does not rescue the altered gene expression seen in NF2-KO cells back to NF2-WT levels.

In sum, we think the most likely explanation is that ME1 is regulated via YAP/TAZ but not via the TEADs. This is a topic we are currently working on, but will probably take an entire project to clarify in a solid way.

- For the 'bottom-up' approach, we performed a bioinformatic enrichment analysis amongst all the genes that are down-regulated in NF2-KO cells compared to NF2-WT cells, to search for transcription factors that regulate this cohort of genes. This analysis identified 17 candidate transcription factors which have the capacity to regulate not only ME1 but a whole panel of down-regulated genes. For instance, amongst them is BACH1, which is known to counteract Nrf2 and thereby repress genes of the oxidative stress response. We then knocked out each transcription factor via CRISPR/Cas9 in NF2-WT and NF2-KO cells and performed Q-RT-PCR on ME1 to see if they abolish the difference in ME1 expression. Interestingly, this large amount of work yielded a handful of transcription factor knockdowns that cause de-repression of ME1 in NF2-KO cells (Reviewer Figure 11).

Reviewer Figure 11: Knockout of 10 different transcription factors leads to de-repression of ME1 and loss of the drop in ME1 expression upon loss of NF2. (A) Fold-change in expression of ME1 in NF2-KO cells compared to NF2-WT cells either in a wildtype background (first column), or in the presence of a knockdown of 17 different transcription factors. Normally, ME1 mRNA levels drop in NF2-KO cells

compared to WT cells (fold-change <0.25). When the transcription factors listed towards the right side (from NR3C1 to GATA3) are knocked out, loss of NF2 no longer causes a drop in ME1. ME1 levels are measured by Q-RT-PCR, normalized to RPL13A.

(B) Sample Q-RT-PCR for ME1 levels in cells with the indicated genotypes.

For instance, when SREBF2 is knocked-out, ME1 levels no longer drop in the NF2-KO compared to the NF2-WT (Reviewer Fig. 11B, bars 3-4). This does not mean SREBF2 is responsible for the drop in ME1 expression upon loss of NF2. To conclude this, we would need to know that SREBF2 activity is increasing in the NF2-KO. It will now require a large amount of work to test individually the activity of all 10 transcription factor hits in NF2-WT and NF2-KO cells to see which ones are changing. We will need to generate luciferase reporter constructs for all 10 transcription factors, and test them in the NF2-WT and NF2-KO cells. Once this is done, and we have identified which transcription factor changes in activity, we will need to understand why. As the reviewer will appreciate, this will constitute an entire study, which goes beyond the scope of a revision.

"Does over-expression of ME1 to rescue the synthetic lethality?"

We now provide data showing this is indeed the case. We transfected the NF2-KO line to express ME1 (Fig. 6C) and this rescues the synthetic lethality caused by pharmacological G6PD inhibition (Fig. 6D).

In sum, we see regulation of ME1 upon loss of NF2 in Schwann cells / Schwannomas in many different genetic backgrounds, indicating it is not an artefact. The molecular connections from NF2 to ME1 likely go via YAP/TAZ but not the via TEADs, to one of the transcription factors shown in Reviewer Fig. 11A, but an additional study will be needed to figure this out. Finally, ME1 levels also drop upon loss of NF2 in patient meningiomas, and the meningioma line we tested was sensitive to G6PD inhibition, indicating that meningioma may be an additional indication worth studying in the future.

Beyond these more fundamental concerns, there are a series of perhaps more minor concerns that nevertheless dampen enthusiasm for the anticipated impact of this work:

1. The abstract over-simplifies the phenotypic consequences of NF2 inactivation, which includes formation of tumors arising from the meningeal lining or ependyma of the CNS. Moreover, it is untrue that "no pharmacological therapy is currently available for patients with NF2, as Avastin/bevacizumab has shown recent promise and approval for treating symptomatic vestibular schwannoma growth in patients with NF2.

We have added to the abstract that NF2 patients also develop meningiomas and ependymomas.

Regarding bevacizumab, we thank the reviewer for pointing this out - we did not intend to be dismissive of the work and progress that has been made in the field. We have reworded the abstract to "few pharmacological options are available for NF2". We also updated the equivalent sentence in the Introduction to read

"There are few pharmacological options for treating NF2 ^{4, 5} although bevacizumab has recently shown promise ⁶."

2. The authors cite the prevalence of G6PD deficiency worldwide in the concluding remarks of their Introduction. What is the prevalence of G6PD deficiency in patients with syndromic NF2? If this is not known or relevant to the study itself, these speculative comments are recommended to be moved to the Discussion.

We have deleted this sentence.

(Please note that the sentence was making no claim whatsoever regarding the possible efficacy of inhibiting G6PD for NF2 therapy, which is what the reviewer is referring to. It simply pointed out that 400 million people worldwide are deficient for G6PD, hence it should be possible to inhibit G6PD without strong side-effects.)

3. Can the authors clarify what other genes have been interrogated and/or identified as altered in ipn0.2 21 cells? This information is important for understanding how specific the reported phenotypes are for syndromic NF2, as opposed to schwannomatosis or sporadic Schwann cell tumors.

As far as we know, there are no known genes that are altered in ipn0.2 21 cells. (This is not a tumor line. These are wildtype, non-tumor cells from a healthy individual that were immortalized.)

The original line is NF2 WT. We generated by CRISPR the NF2-KO mutation to get the isogenic pair. Hence, importantly, the phenotypes we report here are specific for NF2 loss because we are always comparing the isogenic NF2-WT and NF2-KO cells.

4. The blot in Fig. 1a should show a full range of molecular weights to confirm that NF2 has indeed been deleted from ipn0.2 21 cells, using antibodies that recognize either the N- or C-terminal end of the protein to assure complete ablation.

Otherwise, a hypofunctional partial allele may remain. Alternatively, CRISPRi could be used to assure global transcript suppression. Indeed, the noise in the green dots in Fig. 1f suggests that there may be small residual NF2 activity in ipn0.2 2l cells.

We now include as Suppl. Fig. 1B an immunoblot with an additional antibody that recognizes the C-terminus of NF2, including molecular weight markers, which clearly shows that no NF2 protein is left in the knockout cells. (Please note that the genotyping at the DNA level also shows there are no wildtype alleles left.)

5. How specific are the transcriptomic changes in ipn0.2 2l cells with NF2 suppression to the Schwann cell lineage? Choudhury et al. Nat Genet 2022 has reported that meningioma cells with/without NF2 suppression have very different gene expression programs from Schwann/schwannoma cells with/without NF2 suppression. Is this also true for ipn0.2 2l cells? Differential comparison between the data from Choudhury et al. and those reported here are encouraged.

We have compared the gene expression changes caused by loss of NF2 in Schwann cells (our data) versus meningioma cells (Choudhury et al.), and in agreement with the conclusion from Choudhury et al., we see that they are different. We selected all genes found by Choudhury et al. to be differentially expressed upon NF2 re-expression in NF2-KO M10G cells, and plotted for each gene the log₂(FC) in meningioma on the x-axis and log₂(FC) in Schwann cells on the y-axis (Reviewer Figure 12). If the gene expression changes were similar in both cell types, most dots (ie genes) would lie on the y=x diagonal, which is not the case.

Reviewer Figure 12: Loss of NF2 leads to different transcriptional changes in meningioma compared to Schwann cells. Genes found to be differentially expressed in meningioma cells upon loss of NF2 (from Choudhury et al.) are shown. Genes similarly regulated in both cell types would lie on the diagonal.

Meningeal cells and Schwann cells are clearly very different from each other, so the biology in these cells is also different. Since our manuscript is about Schwann cells and not meningioma, we are showing these data to the reviewer here.

6. Why were 7 cell passages used for the genome-wide CRISPR/Cas9 screen?

This is the standard used in the field. In the original paper describing the sgRNA library we used

<https://academic.oup.com/g3journal/article/7/8/2719/6031511>

the authors did 15 doubling times, which is roughly 7 passages.

One needs to wait long enough for the proteins to become depleted, to then yield a proliferation phenotype, to then skew the representations of the sgRNAs in the population. On the other hand, if one waits too long, the data become noisy.

7. The data in Fig. 2a appear to be highly influenced (biased?) by 1 very high value in the NF2-KO#1 control condition. This experiment must be repeated with more replicates to ensure statistical significance (to say nothing of biological significance) are retained. The same is true of Fig. 4d and Fig. 4j

The datapoint that caught the reviewer's attention is from one biological replicate that gave higher values for all the genotypes - ie the cells had proliferated a bit more in all conditions. If anything, this led to a bit more dispersion in the data, reducing the statistical significance. We repeated the experiment and reanalysed the data. The statistical significance and the biological significance are retained.

Fig 4j had 8 biological replicates. Likewise, one of the replicates gave higher numbers for all three genotypes. Since the reviewer considers that replicate biased, we have removed that replicate (which we normally would not do). The fold-change has basically stayed the same (e.g. for KO1 it went up from 42% to 43% compared to the WT), and the difference between WT and KOs has retained its statistical significance. (We usually do not remove any data points because we find it a bit subjective to consider something an outlier.)

We have also added more replicates to Fig. 4d (now Fig 4H), which has also retained its statistical significance.

8. The immunoblot in Fig. 2c does not support the quantification of cell death in Fig. 2b. Namely, there is more cleaved caspase 3 in the NF2-KO#1 control condition of Fig. 2c, but cleaved caspase 3 is increased in the NF2-KO#1 experimental condition of Fig. 2b. This problem appears to also be present in Fig. 2i and Fig. 2j.

There are two issues:

1) Figures 2c and 2b do not show the same thing - one is not a quantification of the other. Figure 2b assesses cell death by CellTox. CellTox is a DNA-binding dye that labels dying cells regardless of their mode of death - this could be apoptosis, ferroptosis, necrosis, etc., not all of which are caspase-dependent. In contrast, Figure 2c shows Caspase 3 cleavage. Hence depending on whether the cells are mainly dying via caspase-dependent apoptosis or not, the two readouts may or may not correlate. The same is true for the respective two panels relating to ACSL3 (now Fig. 3B-C). For instance, in the case of ACSL3, the cells are dying by ferroptosis, so one would not expect the two readouts to correlate. To avoid this misunderstanding, we have now specified on the y-axis of Fig 2b and what is now Fig 3b that these are CellTox assays.

2) We selected to show long exposures of the Caspase 3 immunoblots in the original submission, so that some cleaved-caspase bands would be visible. In absolute terms, however, the amount of caspase cleavage is small - it is not much higher in lanes 2-4, compared to the control condition lane 1 where the cells are happily proliferating without significant cell death. The low amounts of caspase cleavage in all conditions are likely coming from the viral infections. In Suppl. Fig. 3A where we use pharmacological inhibition of G6PD, no cleaved caspase is observed at all.

9. Extended Data Fig. 2 is problematic. The decrease in cell count in EDFig. 2a-b is perhaps 25%, perhaps half of what is shown in the main figure.

This is because the knockdowns of G6PD and ACLS3 by siRNA are less efficient than the corresponding depletions caused by CRISPR/Cas/sgRNA. Hence the magnitude of the effect is different. What is important, however, is that the same synthetic lethality is present using a completely different targeting approach, indicating it is on-target.

There are no error bars or statistics in EDFig. 2d.

We have added data points, error bars and statistics. The result is statistically significant.

Claims of growth factor stimulation with 5% vs 10% serum are suspect and should be repeated in serum-free conditions.

Please note that 10% serum is the normal serum concentration used to culture these cells. The purpose of this experiment was to slow down the cell proliferation by using 5% serum, to test if the synthetic lethality is dependent on the rate of cell proliferation. If we completely remove the serum, the cells die and hence cell proliferation cannot be assayed.

10. The authors report generating xenograft tumors with NF2-KO cells, but do not show images or histology from these reported xenografts. Both are absolutely essential, along with graphs showing tumor initiating capacity and growth rate over time.

We now provide

- images and histology of the tumors in Fig. 2G
- tumor growth curves over time in Figs. 2J, 2K and 3F
- percentages of animals with palpable tumors in Suppl. Fig. 2F

The xenografts carrying the G6PD shRNA in the absence of doxycycline do not grow as robustly as the control tumors with a non-targeting shRNA (Fig. 2K vs 2J) indicating there must be leaky expression of the shRNA in the absence of dox. To be confident of this, we repeated the entire G6PD shRNA xenograft experiment, which took >6 months, and obtained the same outcome. Despite this, the difference + vs - dox is highly statistically significant. The images of the tumors shown in Fig 2G are control tumors, because at the end of the xenograft timecourse, there are no more G6PD knockdown tumors to be found.

Reviewer #3

This study by Kyrkou et al, aims at identifying candidate genes that are synthetic lethal to NF2 mutations to specifically eliminate NF2 mutant cancer cells and spare normal cells.

NF2 encodes for the protein Merlin, a tumor suppressor known to regulate cell-cell adhesion and activate the Hippo pathway through several mechanisms. Merlin is known to be particularly expressed in Schwann cells and to activate the Hypo pathway, MST1/2 and LAM1/2 kinases, and promote the phosphorylation of the cotranscription factors YAP/TAZ and their inactivation/sequestration in the cytoplasm. Neurofibromatosis Type II (NFII) is a rare genetic disorder that can be inherited or can appear as a de novo mutation. In human, mutations in the NF2 gene lead to the production of abnormal shortened version of the Merlin protein and cause Schwannoma. Tumors grow along the vestibulocochlear nerves, a peripheral cranial nerve that controls balance and inner ear functions. Those tumors are usually benign, but cross-sectional studies reveal that if NFII is associated with intracranial meningioma or if diagnosed at young age, NFII becomes life-threatening, and the risk of mortality increases presumably because tumor growth is generally

more rapid in young patients. Because there are no approved chemotherapies for NFII, patients endure surgical removal of the tumors with high risk to lead to nerve damages, post operative complications and death.

Finding therapeutic approach for this very serious life-threatening disease is very significant.

We thank the reviewer for supportive words and pointing out the therapeutic significance of this study.

The authors find one potential target that can specifically reduce cell viability of mutant SC lacking NF2. However, the approach, conclusions, discussion, and technical rigor of this study reduce considerably the enthusiasm to accept this study for publication in nature Communication.

My main problem with this approach is that single targeting synthetic lethal gene without analyzing how Schwann cells compensate for this loss is too risky. The effect obtained might not be the direct consequences of the loss of G6PD but consequential to the dysregulation of other unknown genes when G6PD is lost.

We are perplexed by this comment for two reasons:

1) The concept of synthetic lethality is to target a single gene. In the clinic, this would be done with a specific small-molecule inhibitor. If one needed to target multiple genes, it would be a 'dirty' approach with significant risk of side-effects. So targeting a single target is actually less risky than targeting multiple targets simultaneously.

2) We do indeed understand how the cells compensate for loss of G6PD. We show that the cells produce enough NADPH via the ME1 pathway, and this is the compensatory pathway that enables the cells to be alive. When ME1 is also inhibited, the cells die. The loss of NADPH is indeed a direct consequence of G6PD inhibition, because NADPH synthesis is exactly the molecular function of G6PD. We also show that the synthetic lethality is recapitulated by pharmacologically and acutely blocking G6PD and ME1 where the cells cannot lose other unknown genes.

In addition, while G6PD reduces mutant SC proliferation or increase their cell death, there is a large proportion of mutant SCs still present. Then, the first possible effect of reducing the oxidative stress regulators such as G6PD in the Schwannoma cells is to create more genetic alterations and potentially more aggressive cancerous cells. This point is not discussed and is a big flaw in this study, unless the authors run an RNA-seq in

their NF2-KO SC treated with G6PD shRNA and prove that the treatment is not worse than no treatment.

The reason there are still cells in the dish is because this is only a 4-day assay. With this setup, at 4 days, the control cells become confluent in the dish, so the assay needs to be terminated otherwise it becomes non-linear. To address the reviewer's concern, we now show data where we altered the experimental setup somewhat. We treated the WT and NF2-KO cells with G6PD inhibitor, and then split the cells every 2 days while maintaining the G6PD inhibitor in the medium. This enables the control cells to continue proliferating and not become confluent. After 3 passages one sees a much more dramatic difference, where the NF2-KO cells treated with G6PD inhibitor are essentially all dead (new Fig. 2E-F). This fits with the xenografts, which are also longer-term assays, where we see complete regression of the tumors upon G6PD inhibition. They die completely, and we cannot find any remnant tumors at the end - it certainly does not lead to more aggressive tumors.

Another problem in this study is that while the authors claim that G6PD and ME1 redundantly generate cytosolic NADPH needed by cells to fight oxidative stress and that lack of either one is compatible with cell viability, it is assumed that all Schwann cells will express the same level of ME1. While it is well known that there is a large heterogeneity in SC gene expression and protein production, molecular pathway... there no study showing that all Schwann cells in the body express sufficient level of ME1 to compensate for the lack of G6DP. Thus it is unclear if all non-cancerous Schwann cells won't be affected by a reduction of G6PD expression and generate SC death in other part of the peripheral nervous system.

The fact that 400 million people worldwide are deficient for G6PD and do not have problems with their peripheral nervous system indicates that Schwann cells throughout the body do indeed express enough ME1 to survive.

While the approach of finding a synthetic lethal gene using CRISPR/Cas9 in NF2 mutant SCs is novel, the execution is poor and conclusions made not well supported by the data presented. While the number of single genes in the human CRISPR/Cas9 library being tested is unknown (the authors omitted to give us a number), the fact that a single hit is found among x genes tested might simply mean that the synthetic lethal gene approach is not appropriate for NFII.

The screen was performed using the Toronto CRISPR Human Knockout Library v3, which has 89916 sgRNAs targeting 17232 protein-coding genes. Although this was written in the Materials & Methods, we have now added this to the Results section to make it more clear.

We are not sure why the reviewer thinks the 'execution was poor'. In fact, the screen gave outstanding quality results. The top hits are almost all from the NF2/Hippo pathway. In parallel, we have also been working on other genes that have come out of the screen in the WT cells and they have all re-confirmed upon careful analysis, and we now have several additional interesting projects going on in the lab.

We did find several synthetic lethal genes (see Fig. 1G), but we only focus on the two (not one?) that look most promising. It is unclear to us why the reviewer would expect to find many synthetic lethal genes. To our knowledge, this is not usually the case.

In sum, it is perplexing that the reviewer writes the synthetic lethal approach is not appropriate, if we did indeed find and confirm two genes that are synthetic lethal with NF2.

Also the gene candidate the authors are proposing regulate the oxidative stress cascade, pathway that has already been recently published by the Parkinson's lab in the journal Brain in 2022, that already described pharmacological approach to inhibit this pathway.

Once again, we are perplexed by the reviewer's comment. We assume the reviewer is referring to Laraba et al. Brain 2023 (PMID 36148553). This paper shows that 1) TEAD inhibitors prevents growth of NF2-null schwannomas and meningiomas. This has nothing to do with synthetic lethality - the TEADs are the transcription factors downstream of the NF2-Hippo-YAP pathway and act in an epistatic way.

2) knockdown of ALDH1A1 impairs proliferation of meningioma cells. ALDH1A1 is an aldehyde dehydrogenase. These enzymes catalyze the oxidation of aldehydes to their corresponding carboxylic acids. So they are doing the opposite - they are oxidizing substrates, not reducing them. Furthermore, these results are in meningioma cells, not Schwann cells, which are completely different cells.

The methods used are incomplete and do not enable to reproduce the results.

The authors omitted to describe from which nerves the cells they used in their screening were derived from. It appears that ipn02.3 2λ are derived from human sural nerves. Based on the known heterogeneity of Schwann cell subtypes within the peripheral nervous system, could the authors give a rationale on why using Schwann cells derived from sural nerves (ipn02.3 2λ) is a good model to screen for synthetic lethal gene in NFII, a cranial nerve disease? Are sural nerves affected in NFII patients? If not, the authors need to discuss how molecular regulations of the SCs from sural nerves are comparable to SCs

from the vestibulocochlear nerves (ex: regulation of their oxidative stress level, ME1, AKR1C3, GPX4 expression levels...).

These are the standard cells used by everyone in the community studying NF2. There is, to our knowledge, no NF2-WT immortalized human vestibular Schwann cell line available.

Importantly, the data in Figure 5H, showing that ME1 mRNA levels decrease in NF2 schwannomas compared to control tissue, are from vestibular schwannomas of patients. This indicates that the ipn02.3 2λ line we are using is a good enough model for this purpose because it recapitulates the ME1 regulation observed in patient vestibular schwannomas.

Also worth mentioning is that we see the same regulation of ME1 in ipn02.3 2λ from sural nerve and in the primary human Schwann cells in Suppl. Fig. 4A, which are from the spinal cord, indicating it is not specific to the ipn02.3 2λ line we are using or the sural nerve.

NF2 patients do not only get vestibular schwannomas, but also other schwannomas. This suggests it is a general characteristic of Schwann cells. Indeed NF2 patients can also develop schwannomas of the sural nerve (PMID 9588858). Although originally thought to occur rarely, recent findings suggest that small tumors of peripheral nerves such as the sural nerve happen much more frequently than originally thought (10 out of 15 NF2 patients) when the patients are analyzed carefully (PMID 11960890) resulting in 'sub-clinical' neuropathy. Indeed, mice with NF2 mutations develop schwannomas in almost all locations - uterus, spinal ganglion, skin, submandibular region, retroperitoneum, stomach, small intestine, colon/rectum, bladder, and foreleg (PMID 10887156).

We have now also analyzed data from Ueli Suter's lab where they performed scSeq on different Schwann cells (PMID 33890853). This shows that expression of ME1 is quite constant across different Schwann cell types, even between myelinating and non-myelinating, which are very different (Reviewer Figure 13).

Reviewer Figure 13: ME1
mRNA levels in different Schwann cells of the mouse, from PMID 33890853.

Likewise, NF2 levels are also similar (Reviewer Figure 14).

Reviewer Figure 14: NF2
mRNA levels in different Schwann cells of the mouse, from PMID 33890853.

Other markers, instead, can be quite different (Reviewer Figure 15).

Reviewer Figure 15: Ncam1
mRNA levels in different Schwann cells of the mouse, from PMID 33890853.

In sum, there is no evidence indicating that the biology we are studying is different in Schwann cells at different locations of the body.

We have added to the Results section (when presenting Fig. 5H-I) a sentence explaining that the fact that the gene expression changes are conserved between the sural ipn02.3 2λ line and the patient vestibular schwannomas indicates that the biology is conserved amongst Schwann cells from different locations of the body.

Overall, there is a lack of explanation in the technique used or results obtained that does not help the reader follow the logic of the result interpretation and conclusions.

Here are some examples of lack of clarity:

-It is unclear why both Fig. 1b and 1c are presented, are they both showing the same results, an increase in cell numbers when NF2 is lost? What is Cell Titer Glo compared to Cell Index iCelligence? Are they both indicating cell survival or cell proliferation? material and methods neither legend and results have a clear description about this.

In the section of the M&M where we described these two methods, we now added sentences explaining how each of the two systems measures cell number. The Cell Titer Glo system works well for low cell numbers, but becomes saturated when cells become confluent in the well. For this reason we used the iCELLigence for the measurements at higher cell density. The main message is that at sub-confluence, WT and NF2-KO cells proliferate at roughly the same rate (Fig. 1B), but when WT cells become confluent and become contact inhibited, the NF2-KO cells continue proliferating (Fig. 1C).

-Why the relative cell number is lower in NF2-KO/Cas9 versus NF2-KO#1 and NF2-KO#2? Performing a CRISPR/cas9 for NF2 in NF2-KO mutant cells should not impact cell number unless the CRISPR/cas9 for NF2 has also off targets. The authors need to provide an explanation for this discrepancy.

There seems to be a misunderstanding here. The NF2-KO/Cas9 cells only differ from the NF2-KO#1 cells in that they express Cas9. The NF2-KO/Cas9 line does not have any sgRNA in it. This is not a "CRISPR/cas9 for NF2" because there is no sgRNA.

Cell lines often grow more slowly when they overexpress a protein.

- 'This identified many genes required for cell viability in both cell lines ("essential genes", Fig. 1f).'

It is unclear if the following sentence means that all the other genes presented in Fig1f are essential genes except for G6PD and ACSL3.

Figure 1f shows the proliferative effect of knocking down every single gene in the genome in NF2-WT (x-axis) versus NF2-KO cells (y-axis). The $\log_2(\text{fold change})$ is being plotted, meaning that positive numbers indicate faster proliferation, and negative numbers slower proliferation. The dots that show strongly negative numbers in both WT and KO cells (ie the dots in the lower left quadrant of the graph) are the essential genes required for viability. We've now added a box to Figure 1f to make more clear which genes are the essential ones.

- For the genome-wide CRISPR/Cas9 library how many gRNAs targeting how many protein coding genes, and which control were used to control for the rate of infections? It is important to discuss the number of positive hits (2?) versus 'x' number of targets tested, this could mean that NF2 mutant Schwann cells have low level of sensitivity to synthetic lethal gene approach.

We used the Toronto CRISPR Human Knockout Library v3, which has 89916 sgRNAs targeting 17232 protein-coding genes.

As far as we can tell, the Reviewer is suggesting that the more synthetic lethal genes one finds, the better, because it indicates the cell line is "sensitivity to synthetic lethality". We politely disagree with the reviewer - we think this point of view is not correct:

- As far as we know, it is not really possible for different cell lines to have different 'sensitivity to synthetic lethality'. Synthetic lethality is the concept that if two genes are redundantly performing one function required for cell viability, then deleting both genes causes the cell to die. The only way for a cell line to be more or less sensitive to 'synthetic lethality' as a general phenomenon is if:

- 1) The cell has lost hundreds of genes causing a reduction in the number of redundant gene pairs. In this case, the synthetic lethality is gone because deleting the one remaining gene for each function is already sufficient to kill the cell.

- 2) The cell has gained hundreds of redundant gene pairs that are not present in other cells of the same organism. In this case, deleting the redundant gene pair would not cause synthetic lethality because additional redundant genes performing the same function have been gained. This would likely require a genome duplication and evolution, which does not happen within one organism.

- The number of synthetic lethal genes that one finds will depend instead on what gene being tested. Typically, the synthetic lethal genes are the ones performing redundant functions with the one being tested.

- If a gene A performs a unique function in a cell that is not redundant with anything else, then loss of gene A will be lethal and 'synthetic lethality' does not apply.
- At the other end of the spectrum, if gene A performs a function that is extremely redundant, with lots of other genes compensating for gene A, then again one will not find synthetic lethality, because one would need to simultaneously inactivate all the other compensating genes to obtain lethality, which is experimentally unlikely.
- However, if the situation is somewhere in between, and gene A does something redundant with one or two other genes, then there's a chance to find these one or two other genes.

In sum, we believe the number of synthetic lethals that one finds is a property of the gene being assayed, not a general property of the cell line.

•We also disagree that finding more synthetic lethals would be a better outcome. If anything, it would suggest that no matter how you 'touch' the cell, it dies, raising concern that the cell line is very unhealthy and that the synthetic lethality that one sees is actually just a non-specific sensitivity to any perturbation. So if anything, finding few synthetic lethals for a gene suggests that one has found something specific. A priori, one would only expect to find a handful of genes, which is what we found here.

Regarding rates of infection: For CRISPR/Cas9 screening, the cells are transduced with virus at an MOI of less than 1 (in our case 0.3) so that cells only get 1 sgRNA per cell. The transduced plasmid contains a puromycin resistance cassette, so that after transduction the cells are treated with puromycin to only select the cells that were successfully transduced. (This is described in the Materials & Methods.)

- 'We excluded from further consideration any gene that causes increased proliferation of NF2-WT cells, since these are potentially tumor suppressors and could lead to tumors if targeted pharmacologically in patients. Likewise, we excluded any gene that causes reduced cell numbers in NF2-WT cells since targeting them pharmacologically could be toxic.'

Where is the list of those genes?

The list of genes is in Figure 1G, as mentioned in the sentence directly before:

"Figure 1G shows the top genes sorted by their differential effect on NF2-KO versus NF2-WT cells. We excluded from further consideration any gene that causes ..."

- 'NF2-WT and NF2-KO cells at low density so that both lines showed a similar ~5-fold increase in cell number over the course of 4 days when transduced with negative-control sgRNA (Fig. 2a).'

Please clarify what was used as negative control sgRNA and how the infection efficiency was controlled.

The negative control sgRNA targets AAVS1. This is a standard one used in the CRISPR/Cas field. (This was described in the sentence directly before the one cited by the reviewer).

Cells were selected after transduction with puromycin, so only transduced cells are used in the assay.

- 'This was due at least in part to a significant, 6-fold increase in cell death upon G6PD loss-of-function in NF2-KO cells, assessed by CellTox (Fig. 2b).'

It is unclear if the Cell Tox was also done after 4 days in culture so that the corresponding reduce number of cells in NF2KO G6PD Fig2a can really be correlated with the 6 fold cell death fig 2b.

Yes - Cell Tox was also assayed at Day 4, directly in parallel with the viability measurements. We have added this to the y-axis label of the figures.

Concerning the xenograft experiment

-Because there is no explanation why RLuc is used as a negative control and there is no quantification of luciferase expression it is unclear what the control RLuc is used for in Figure2? please clarify.

The cells do not express RLuc, so RLuc shRNA is simply used as a non-targeting negative-control shRNA. We have now clarified this in the text.

-There are no representative images of the cells post subcutaneous xenograft that have been used for all the quantifications presented in Figure2. It is unclear how those quantifications were done in Figure 1 and 2, please also provide a detailed explanation of the methods used for quantifications (ex: just citing iCelligence or Cell Titer Glo is not sufficient).

We now added as Fig. 2G both macroscopic images of the excised tumors, and histological sections. (Please note that the G6PD tumors all die by the end of the xenograft timecourse, so it's only possible to image the control tumors.)

Tumor volume was measured with a caliper up to three times a week and calculated according to the formula: $V = (\text{length (mm)} \times \text{width (mm)}^2)/2$. This is mentioned in the Materials & Methods in the section " Schwann cell cancer xenograft experiment".

-It is unclear why there is investigation on why the loss of ACSL3 lead to NF2-KO SC death while why the mechanism of G6PD loss of expression that lead SC death is not being explored while at the end of the study the authors clearly mention that the more targetable gene to treat NFII is G6PD. "Overall, to us G6PD appears most promising because a G6PD inhibitor is likely to have a mild side-effect profile if dosed appropriately".

We are perplexed by this comment. We do indeed study why loss of G6PD causes death - in fact we study this more in depth than we study the mechanism of ACSL3-induced cell death ? In the original submission, Figure 3 focused on understanding how ACSL3 knockout causes cell death, while Figure 4 focused on understanding how G6PD inhibition causes cell death. We show that G6PD and ME1 are redundant for generating NADPH. When one enzyme or the other is inhibited, the Schwann cells can cope with the NADPH produced by the other source, thereby maintaining glutathione sufficiently reduced. When both are inhibited, however, cells become oxidized and die. We show that loss of NF2 leads to increased oxidation in Schwann cells, in part due to reduced ME1 levels. Perhaps the reviewer missed Figure 4 ?

(We have now re-organized the manuscript to highlight what is in common between ACSL3 and G6PD. Hence, the organization of the figures has changed, but the manuscript still contains more mechanistic investigation on G6PD than ACSL3.)

It is not clear why a subcutaneous xenograft of the NF2-KO cells was used instead of onto the cochleovestibular nerve, knowing that the Hippo pathway and mechanotransduction respond differently depending on the cellular environment (abdomen cavity vs cochleovestibular region).

To our knowledge, there is no established method for orthotopic NF2 tumors into the cochleovestibular nerve of mice. Indeed, we also had to set up the subcutaneous model, because this one also did not exist prior to our study. We think the subcutaneous model is also appropriate, because Schwannomas also develop subcutaneously in NF2 patients.

The manuscript has big flaws and misses the opportunity to cite key publications in the field of Schwannoma cells Hippo pathway and YAP TAZ. Consequently, inaccurate conclusions are made. For example, saying that YAP and TAZ are required for myelin maintenance is wrong since this is still debated in the field (Grove versus Jeannette et al, 2020).

The Reviewer claims we did not cite key publications in the field, but the reviewer may have missed that we did cite in our original submission both papers the reviewer mentions here - Grove et al. 2017 (it was citation #9) and Jeanette et al. 2021 (it was citation #7).

To our understanding, Grove et al. find that YAP/TAZ are required for myelin maintenance, while Jeanette et al. find that they are not required for maintenance, but for repair after injury. We have therefore reworded the sentence to read:

"This approach is very promising, although toxicity may arise, given that genetic loss of YAP/TAZ may lead to impaired maintenance of Schwann cell myelination or to impaired remyelination after injury, and consequently impaired nerve function^{9, 10, 11}."

The authors never use Merlin in their manuscript, which is the protein coded by NF2.

NF2/Merlin is one of the few examples in the human genome where the gene name and the gene product / protein name don't match. This is for historical reasons, because Merlin is the name of the Drosophila gene (and protein) which was cloned/identified prior to the human one. Indeed, half of our lab works with Drosophila and we are well aware of this history. The problem is that it would be extremely confusing if the 20,000 genes in our genome had 20,000 different names for their protein gene products. Also, it is standard in the molecular biology field to name gene products with the same name as the gene. Hence, to make our findings more accessible to a broad audience, we decided to follow the standard in the field.

e.g. see Protein atlas: <https://www.proteinatlas.org/ENSG00000186575-NF2> where the protein name is provided as either NF2 or Merlin.

Overall, this shows a lack of rigor in preparing the manuscript and lack of knowledge of the field.

Schultz et al., 2013, Mindos et al., 2017, Jeannette et al, 2020, Feltri et al., 2021, Labara et al., 2022 ...

We are sorry the reviewer was frustrated by the fact that we did not cite these papers. This was not intentional.

Schultz et al., 2013: we assume the reviewer means Schulz et al. 2013, PMID 23455610 ? (We did not find any relevant papers from Schultz - with a 't'). As far as we understand, this paper shows that NF2 not only has a role in Schwann cells, but also has a function in neurons to maintain axonal integrity, with lack of NF2 thereby leading to polyneuropathy. We did not cite this, because our paper focuses on Schwann cells, so this does not seem directly relevant. Please let us know if the Reviewer meant a different paper.

Mindos et al., 2017: we have added a citation to this.

Jeannette et al, 2020: Jeanette et al 2021 was already cited in our original submission. We assume this is the paper the reviewer means?

Feltri et al., 2021: we found three papers from 2021 with Feltri as first author, one of which is a review on Hippo signaling - PMID 33449435. We have added a citation to this.

Labara et al., 2022: we cannot find any paper in PubMed with Labara as an author. We found some papers for Labarca 2022 but they all relate to COVID and sleep apnea. We assume the reviewer means Laraba et al. 2023? We have added a citation to this paper.

REVIEWER COMMENTS

Reviewer #1 (Remarks to the Author):

The authors have satisfactorily answered my questions.

Reviewer #2 (Remarks to the Author):

The authors have undertaken a major revision of their study, which is largely improved.

Rationale for targeting G6PD in cells with heterozygous vs homozygous NF2 mutations: In response, the authors have generated a "Schwann cell line" with inactivation of one copy of NF2 and a single amino acid missing in the second copy of NF2 (Fig. S2H) and demonstrate that genomic editing has reduced the expression of Merlin protein (Fig. S2I) but not impacted the proliferation of these cells (Fig. S2J). They also generated non-Schwann/glia NF2 knockout cells and showed a lack of synthetic lethality with G6PD inhibition (Fig. S7). These experiments are very much appreciated, but the identity of the Schwann cell line must be disclosed and verified to confirm that these cells are indeed normal/wildtype.

Rationale for highlighting ACSL3 in the narrative: In response, the authors have altered the title of their study re-ordered/re-written their manuscript to highlight the commonalities with G6PD and ACSL3. These textual edits are appreciated.

Observation that ME1 is suppressed in Schwann cells: In response, the authors cite ample cell line (Schwann and non-Schwann in origin) and patient data to support this finding, and now incorporate data from HEI-193 syndromic schwannoma cells, meningioma cells, myriad data looking at the transcriptional control of ME1, and rescue experiments that support their hypothesis. These data are robust and impressive, but it appears that much of these data are limited to only the response letter and not the manuscript itself. These data should be incorporated into the supplemental data of the manuscript, at the very least. This is relevant to all "Reviewer Figures."

Initially minor concerns: Most responses are adequate, but now it appears the textual responses have not permeated the revised manuscript itself. This should be corrected, and the additional text clarifying experimental conditions and findings should be incorporated into the Methods, at the very least. The one area of now significant concern is the authors speculative claim that leaky anti-G6PD shRNA expression from their dox-promoter is significantly inhibiting the growth of control tumors in Fig. 2K. If anything, it looks like tumors in both conditions of this graph failed to grow over the 165 days they were culture in vivo (i.e. the tumor sizes at day 17 are essentially identical to the tumor sizes at day 165). How can this be? The authors must show histology from both conditions (anti-G6PD shRNA +/- dox) so that the readers can confirm the tumors in this experiment are indeed cellular masses (the tumor shown in Fig. 2G appears largely acellular/reactive) and immunoblots from both conditions relative to the RLuc-shRNA conditions +/- dox to shed light on their speculation that shRNA expression must be leaky in the absence of dox. In the absence of such data, it is equally likely that no tumors truly formed in the G6PD-shRNA conditions (with or without dox) and all changes measures were merely reactive.

Reviewer #3 (Remarks to the Author):

Based on the information provided, it appears that most of my concerns raised have been addressed by the authors in their responses. The authors have provided explanations, data, and clarifications to respond to the specific points raised.

Reviewer #1

The authors have satisfactorily answered my questions.

We thank the reviewer for the helpful suggestions and the positive evaluation.

Reviewer #2

The authors have undertaken a major revision of their study, which is largely improved.

Rationale for targeting G6PD in cells with heterozygous vs homozygous NF2 mutations: In response, the authors have generated a "Schwann cell line" with inactivation of one copy of NF2 and a single amino acid missing in the second copy of NF2 (Fig. S2H) and demonstrate that genomic editing has reduced the expression of Merlin protein (Fig. S2I) but not impacted the proliferation of these cells (Fig. S2J). They also generated non-Schwann/glia NF2 knockout cells and showed a lack of synthetic lethality with G6PD inhibition (Fig. S7). These experiments are very much appreciated, but the identity of the Schwann cell line must be disclosed and verified to confirm that these cells are indeed normal/wildtype.

We apologize for not having described this more clearly in the text. We generated the NF2 heterozygous Schwann cell line using CRISPR/Cas9 on the very same wildtype Schwann cells (ipn02.3 2λ) and using the same sgRNA as we did to generate the NF2 homozygous knockout line, except this time we screened clones for heterozygous loss of NF2 rather than homozygous loss of NF2. As a result, we have three Schwann cell lines that are derived from the same line, and hence are isogenic except for NF2 being either +/+, +/- or -/-. This makes the results of the G6PD inhibition experiment directly comparable between the three lines. We have clarified this in the text:

"To test this, we again CRISPRized the ipn02.3 2λ wildtype Schwann cell line with the same sgRNAs that we used to generate the NF2-KO cells, except this time we screened for heterozygous loss of NF2 rather than homozygous loss. This yielded a Schwann cell line that is isogenic to ipn02.3 2λ except it has a frame-shift mutation and a premature stop codon on one NF2 allele (Suppl. Fig. 2H)."

Rationale for highlighting ACSL3 in the narrative: In response, the authors have altered the title of their study re-ordered/re-written their manuscript to highlight the commonalities with G6PD and ACSL3. These textual edits are appreciated.

Thanks.

Observation that ME1 is suppressed in Schwann cells: In response, the authors cite ample cell line (Schwann and non-Schwann in origin) and patient data to support this finding, and now incorporate data from HEI-193 syndromic schwannoma cells, meningioma cells, myriad data looking at the transcriptional control of ME1, and rescue experiments that support their hypothesis. These data are robust and impressive, but it appears that much of these data are limited to only the response letter and not the manuscript itself. These data should be incorporated into the supplemental data of the manuscript, at the very least. This is relevant to all "Reviewer Figures."

We actually put most of the data directly into the manuscript:

- Suppl. Fig. 7 (now Suppl. Fig. 8) contains the lines of non-Schwann cell origin.
- Fig. 5 and Suppl. Fig. 4 contain the patient data.
- Suppl. Fig. 4 and Suppl. Fig. 6 (now Suppl. Fig. 7) contain the HEI-193 data.
- Fig. 6C-D contain the rescue data.

As suggested by the reviewer, we have now shifted Reviewer Figures 1 and 9 (which were the same) into Supplemental Figure 6A, Reviewer Figures 2 and 10 (which were the same) into Supplemental Figures 6B-C, and Reviewer Figures 5-6 into Supplemental Figure 6D-E.

We did not incorporate into the manuscript the IOMM-Lee and the rat Schwann cell data because they are not useful. The data we presented to the reviewers show that loss of NF2 does not lead to activation of YAP in these two cell lines, so they are not useful models to study NF2. The data on the other transcription factors (Reviewer Figure 11) are too preliminary to put into the manuscript.

Initially minor concerns: Most responses are adequate, but now it appears the textual responses have not permeated the revised manuscript itself. This should be corrected, and the additional text clarifying experimental conditions and findings should be incorporated into the Methods, at the very least.

With incorporation of the new data indicated above, along with associated text, we believe the textual responses to the reviewers are reflected in the manuscript. If there is still a specific point which the reviewer feels was addressed in the Response to Reviewers but not incorporated in the manuscript, please let us know and we're happy to do so.

The one area of now significant concern is the authors speculative claim that leaky anti-G6PD shRNA expression from their dox-promoter is significantly inhibition the growth of control tumors in Fig. 2K. If anything, it looks like tumors in both conditions of this graph failed to grow over the 165 days

they were culture in vivo (i.e. the tumor sizes at day 17 are essentially identical to the tumor sizes at day 165). How can this be? The authors must show histology from both conditions (anti-G6PD shRNA +/- dox) so that the readers can confirm the tumors in this experiment are indeed cellular masses (the tumor shown in Fig. 2G appears largely acellular/reactive) and immunoblots from both conditions relative to the RLuc-shRNA conditions +/- dox to shed light on their speculation that shRNA expression must be leaky in the absence of dox. In the absence of such data, it is equally likely that no tumors truly formed in the G6PD-shRNA conditions (with or without dox) and all changes measures were merely reactive.

We now provide data clearly showing that our hypothesis was correct - the reason the G6PD shRNA -dox tumors did not grow as large as the negative control RLuc-shRNA tumors was due to leakiness of the shRNA. We did an immunoblot on three RLuc-shRNA -dox control tumors and three G6PD shRNA -dox tumors and found that G6PD protein levels are significantly reduced in the G6PD shRNA -dox tumors (Suppl. Fig. 2G).

The tumors are not reactive, as can be judged by the growth dynamics. In this xenotransplantation model, the matrigel/cell pellets are no longer visible 2 weeks after injection. After another 5 weeks, palpable tumors grow out with a measurable diameter of 0.3-0.5 mm.

The tumors are also not acellular. We now provide in Suppl. Fig. 2G a higher magnification image of the periphery of a G6PD-shRNA tumor -dox, showing nuclei that are elongated, appearing in spindle form and in comma shape, typical of Schwannomas. Only the core of the tumor is necrotic. One of the authors on this paper is the head of the Xenograft Core Facility of the German Cancer Research Center since more than a decade, and from her experience, necrotic cores are very common in xenografts. Indeed, the necrotic core shown in Fig 2G is from a control RLuc-shRNA -dox tumor that grew large (Fig. 2J), hence the presence of a necrotic core does not coincide with a small tumor.

Please note we cannot show histology on the G6PD-shRNA tumors +dox because they are all dead and non-existent at the end of the xenograft timecourse.

Reviewer #3

Based on the information provided, it appears that most of my concerns raised have been addressed by the authors in their responses. The authors have provided explanations, data, and clarifications to respond to the specific points raised.

We thank the reviewer for the helpful suggestions and the positive evaluation.

REVIEWERS' COMMENTS

Reviewer #2 (Remarks to the Author):

Nice work! Thank you for your clarifications and additions.